# Functional profiling of long intergenic non-coding RNAs in fission yeast

Maria Rodriguez-Lopez[1], Shajahan Anver[1†], Cristina Cotobal[1†],
Stephan Kamrad[1,2,3†], Michal Malecki[1†‡], Clara Correia-Melo[2], Mimoza Hoti[1],
StJohn Townsend[1,2], Samuel Marguerat[1§], Sheng Kai Pong[1], Mary Y Wu[4],
Luis Montemayor[1], Michael Howell[4], Markus Ralser[2,3], Jürg Bähler[1*]

[1]University College London, Institute of Healthy Ageing and Department of Genetics, Evolution & Environment, London, United Kingdom; [2]The Francis Crick Institute, Molecular Biology of Metabolism Laboratory, London, United Kingdom; [3]Charité Universitätsmedizin Berlin, Institute of Biochemistry, Berlin, Germany; [4]The Francis Crick Institute, High Throughput Screening, London, United Kingdom

**\*For correspondence:**
j.bahler@ucl.ac.uk

[†]These authors contributed equally to this work

**Present address:** [‡]Institute of Genetics and Biotechnology, Faculty of Biology, University of Warsaw, Warsaw, Poland; [§]UCL Cancer Institute, Genomics Translational Technology Platform, London, United Kingdom

**Competing interest:** The authors declare that no competing interests exist.

**Abstract** Eukaryotic genomes express numerous long intergenic non-coding RNAs (lincRNAs) that do not overlap any coding genes. Some lincRNAs function in various aspects of gene regulation, but it is not clear in general to what extent lincRNAs contribute to the information flow from genotype to phenotype. To explore this question, we systematically analysed cellular roles of lincRNAs in *Schizosaccharomyces pombe*. Using seamless CRISPR/Cas9-based genome editing, we deleted 141 lincRNA genes to broadly phenotype these mutants, together with 238 diverse coding-gene mutants for functional context. We applied high-throughput colony-based assays to determine mutant growth and viability in benign conditions and in response to 145 different nutrient, drug, and stress conditions. These analyses uncovered phenotypes for 47.5% of the lincRNAs and 96% of the protein-coding genes. For 110 lincRNA mutants, we also performed high-throughput microscopy and flow cytometry assays, linking 37% of these lincRNAs with cell-size and/or cell-cycle control. With all assays combined, we detected phenotypes for 84 (59.6%) of all lincRNA deletion mutants tested. For complementary functional inference, we analysed colony growth of strains ectopically overexpressing 113 lincRNA genes under 47 different conditions. Of these overexpression strains, 102 (90.3%) showed altered growth under certain conditions. Clustering analyses provided further functional clues and relationships for some of the lincRNAs. These rich phenomics datasets associate lincRNA mutants with hundreds of phenotypes, indicating that most of the lincRNAs analysed exert cellular functions in specific environmental or physiological contexts. This study provides groundwork to further dissect the roles of these lincRNAs in the relevant conditions.

## Introduction

Genomes produce pervasive and diverse non-coding RNAs. How much genetic information is transacted by this non-coding 'dark matter' remains a matter of debate. A substantial but poorly understood portion of transcriptomes consists of long intergenic non-coding RNAs (lincRNAs). lincRNAs are longer than 200 nucleotides, lack long open reading frames, and do not overlap any neighbouring coding regions. While not all lincRNAs may be functional, several have well-defined roles in gene regulation and some other cellular processes. Different lincRNAs can control gene expression at different levels, from transcription to translation, and either in *cis* (acting on neighbouring genes) or in *trans* (acting on distant genes) (*Fauquenoy et al., 2018*; *Popadin et al., 2013*; *Rinn and Chang, 2012*; *Schlackow et al., 2017*; *Ulitsky and Bartel, 2013*; *Yamashita et al., 2016*). Although lincRNAs show little sequence conservation between species, functional principles seem to be conserved which

can help us to understand their biology (*Ulitsky, 2016*). Specific lincRNAs have been implicated in complex human diseases (*Batista and Chang, 2013 Kumar et al., 2013*). For example, Xist exerts a tumour-suppressive function (*Yildirim et al., 2013*), TUNA is associated with neurological function and Huntington's disease (*Lin et al., 2014*), and lincRNA1 delays senescence (*Abdelmohsen et al., 2013*). Moreover, lincRNAs are emerging as diagnostic molecular markers as they can be easily detected in blood and could provide more readily accessible drug targets than proteins (*Bester et al., 2018*; *DeWeerdt, 2019*; *Kim et al., 2016*).

Despite these efforts and insights based on studying selected lincRNAs, the systematic picture remains incomplete as the importance of most lincRNAs is unknown. Functional analyses of lincRNAs are challenging given their profusion, poor annotation, low expression, and limited methodology (*Bassett et al., 2014*; *Cao et al., 2018*; *Kopp and Mendell, 2018*). Knowledge of lincRNA function is therefore scarce even in well-studied organisms, highlighting the need for more systematic approaches. Large-scale genetic studies of lincRNAs and other non-coding RNAs have emerged, starting to provide a more global picture on their functions and contributions to phenotypes (*Balarezo-Cisneros et al., 2021*; *Bester et al., 2018*; *Huber et al., 2016*; *Joung et al., 2017*; *Liu et al., 2017*; *Parker et al., 2018*; *Tuck et al., 2018*; *Wei et al., 2019*). These findings suggest that many lincRNAs play specialized roles in specific conditions and, therefore, need to be analysed in the relevant conditions.

The fission yeast, *Schizosaccharomyces pombe*, is a potent genetic model system to study gene regulation and lincRNA function in vivo (*Atkinson et al., 2018*; *Fauquenoy et al., 2018*; *Marguerat et al., 2012*; *Yamashita et al., 2016*). Although only the most highly expressed lincRNAs show purifying selection (*Jeffares et al., 2015*), their regulation is often affected by expression quantitative trait loci (*Clément-Ziza et al., 2014*). Notably, transposon insertions in up to 80% of non-coding regions of the *S. pombe* genome can affect fitness (*Grech et al., 2019*). RNA metabolism of fission yeast is similar to metazoan cells. For example, RNA interference (RNAi), RNA uridylation, and PABPN1-dependent RNA degradation are conserved from fission yeast to humans, but absent in budding yeast. Genome-wide approaches by us and others have uncovered widespread lincRNAs in fission yeast (*Atkinson et al., 2018*; *Eser et al., 2016*; *Rhind et al., 2011*; *Wilhelm et al., 2008*). Nearly all *S. pombe* lincRNAs are polyadenylated and transcribed by RNA polymerase II (*Marguerat et al., 2012*). Transcription of lincRNAs starts from nucleosome-depleted regions upstream of positioned nucleosomes (*Atkinson et al., 2018*; *Marguerat et al., 2012*), and the regulation of some lincRNAs involves specific transcription factors such as Gaf1 (*Rodríguez-López et al., 2020*). Most *S. pombe* lincRNAs are cryptic in cells growing under standard laboratory conditions, being suppressed by RNA-processing pathways such as the nuclear exosome, cytoplasmic exonuclease, and/or RNAi (*Atkinson et al., 2018*; *Zhou et al., 2015*), but they become induced during starvation or sexual differentiation (*Atkinson et al., 2018*). A substantial portion of lincRNAs are actively translated (*Duncan and Mata, 2014*), raising the possibility that some of them code for small proteins. A few *S. pombe* lincRNAs have been functionally analysed: *meiRNA, mamRNA, nam1,* and *rse1* control meiotic differentiation (*Andric et al., 2021*; *Ding et al., 2012*; *Fauquenoy et al., 2018*; *Touat-Todeschini et al., 2017*; *Yamashita et al., 2016*), *SPNCRNA.1164* regulates the *atf1* transcription factor gene in *trans* during oxidative stress (*Leong et al., 2014*), several lincRNAs activate the downstream *fbp1* gene during glucose starvation (*Oda et al., 2015*), *prt* controls *pho1* expression (*Ard et al., 2014*; *Shah et al., 2014*), and *nc-tgp1* inhibits the *tgp1* gene by transcriptional interference (*Ard et al., 2014*).

Most *S. pombe* lincRNAs may not function under benign laboratory conditions when they are typically very lowly expressed (*Atkinson et al., 2018*; *Marguerat et al., 2012*). Phenomics approaches seek to rigorously characterize phenotypes associated with many gene variants under diverse conditions (*Brochado and Typas, 2013*; *Rallis and Bähler, 2016*). Such broad, high-throughput (HTP) phenotyping is an effective approach to uncover functional clues for unknown genes. For example, while only 34% of all budding yeast gene-deletion mutants display a growth phenotype under the standard condition, 97% of these mutants show suboptimal growth in at least one condition when assayed under a large number of chemical or environmental perturbations (*Hillenmeyer et al., 2008*). We have established a sensitive, reproducible platform for HTP colony-based assays to determine cellular fitness under diverse conditions (*Kamrad et al., 2020b*). Here we take advantage of this potent approach to broadly investigate phenotypes of 150 lincRNAs (12.6% of the 1189 lincRNAs; *Atkinson et al., 2018*), using deletion and/or overexpression mutants, supplemented with HTP microscopy and flow cytometry assays of deletion mutants. Colonies of a representative set of 238

coding-gene mutants were phenotyped in parallel for functional comparison. Using these different assays, we collected quantitative data for over 1.1 million unique colonies and over 5.7 million cells in a wide range of conditions. This study reveals hundreds of novel lincRNA-associated phenotypes and provides a framework for follow-on studies.

## Results and discussion
### Experimental strategy for functional profiling of lincRNAs

We focused on lincRNAs, rather than other types of non-coding RNAs, because (1) they are poorly characterized in general but emerge as varied regulatory factors; (2) they can be deleted without directly interfering with coding gene function; and (3) they are more likely to function in *trans* as RNAs than antisense or promoter-associated ncRNAs which can affect neighbouring or overlapping genes via their transcription (*Ard et al., 2017*). For functional profiling, we selected 150 *S. pombe* lincRNA genes that produce well-defined transcripts and are well-separated from neighbouring coding regions (over ~200 bp), based on genome browser views of RNA-seq data. We established efficient HTP methods to genetically manipulate these lincRNAs. For deletions, we applied a CRISPR/Cas9-based method (*Rodríguez-López et al., 2016*); this approach allowed us to knock out the precise regions transcribed into lincRNAs without inserting any markers or other alterations, thus avoiding indirect physiological effects. For overexpression, we applied restriction-free cloning to express the lincRNAs from a plasmid under the control of the strong, inducible *nmt1* promoter (*Maundrell, 1993*). Gene overexpression ('gain of function') provides complementary phenotype information to gene deletion (*Prelich, 2012*); moreover, any phenotype caused by a lincRNA that is ectopically expressed from a plasmid points to a function that is exerted over a distance (in *trans*) via the lincRNA itself rather than via its transcription or other local effects. We managed to delete 141 different lincRNAs (111 of which with at least two independent guide RNAs) and to overexpress 113 lincRNAs, with 104 lincRNAs being both deleted and overexpressed. These lincRNAs ranged in length from ~90 to 5100 nucleotides and in GC content from 25% to 46%, with means of 820 nucleotides and 34% GC content, respectively. These lincRNAs are distributed across the entire nuclear genome (*Figure 1A*). Information for all deletion and overexpression strains analysed is available in *Supplementary file 1*.

To provide functional context for the lincRNA deletion-mutant phenotypes, we also assayed 238 coding-gene mutants from the *S. pombe* gene-deletion library using prototrophic mutants after crossing out the auxotrophic mutants (*Malecki and Bähler, 2016*). These mutants broadly cover the Gene Ontology (GO) slim Biological Process categories (*Lock et al., 2019*), ageing-related genes (*Rallis et al., 2014*; *Sideri et al., 2014*), as well as 104 'priority unstudied genes' (*Wood et al., 2019*; *Supplementary file 1*).

*Figure 1B* provides an overview of the colony- and microscopy-based phenomics assays for the deletion and overexpression mutants. To determine fitness-related traits from colony-based assays, we applied *pyphe,* our Python package for phenomics analyses (*Kamrad et al., 2020b*). Strains were arrayed randomly around a control grid at a density of 384 colonies per plate. We assayed the deletion or overexpression strains in response to diverse environmental factors such as different nutrients and drugs as well as oxidative, osmotic, heavy-metal, protein-homeostasis and DNA-metabolism stresses (*Figure 1—figure supplement 1*), including some combined factors which can reveal additional phenotypes by non-additive effects that are not evident from single conditions (*Rallis et al., 2013*). For drugs and stressors, we applied low and high doses, where wild-type cell growth is normal or inhibited, respectively, to uncover both sensitive or resistant mutants. For the deletion mutants, we measured colony size to determine cell growth across 149 different nutrient, drug, and stress conditions (*Supplementary file 1*). For 68 of these conditions, we also measured colony redness using the phloxine B dye, which stains dead cells and thus provides a measure for viable cells in colonies: bright red colonies have higher portions of viable cells than dark red colonies (*Kamrad et al., 2020b*; *Lie et al., 2018*). Cell growth and viability provided complementary functional information and produced strong biological signals (*Figure 1C*). For the overexpression mutants, we assayed cell growth across 47 conditions (*Supplementary file 1*). All colony-based phenotyping was performed in at least three independent biological repeats per condition and strain, with a median number of nine independent repeats per lincRNA, and at least two technical repeats (independently printed colony) for each biological repeat.

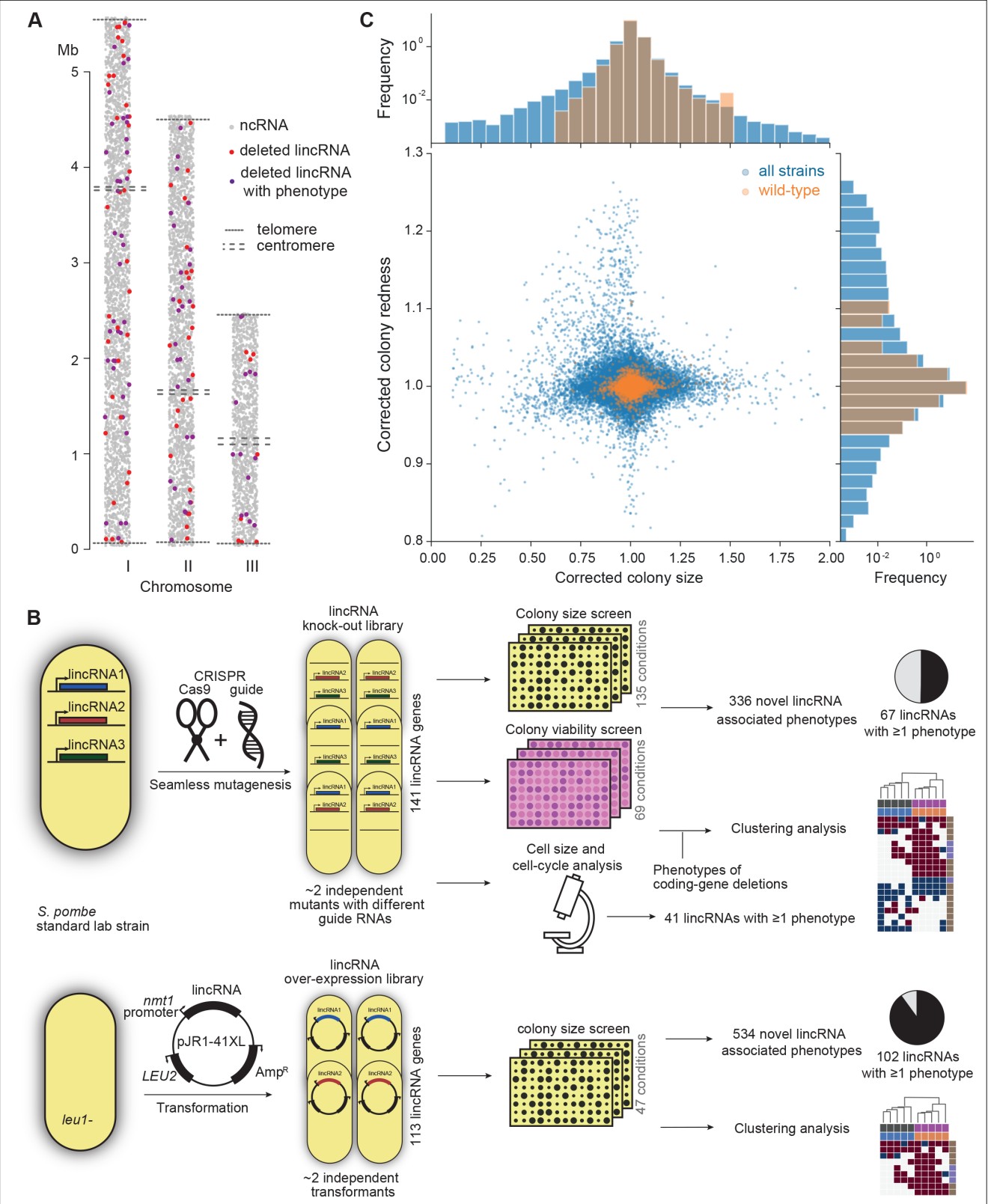

**Figure 1.** Phenomics assays of long intergenic non-coding RNA (lincRNA) mutants. (**A**) Representation of all non-coding RNAs across the three *S. pombe* chromosomes (*Atkinson et al., 2018*). lincRNAs analysed in this study are shown in red (if they showed no phenotypes) or purple (if they showed phenotypes in at least one condition), with all other non-coding RNAs in grey. (**B**) Schematic overview of experimental design and workflow for phenotyping and data analyses. (**C**) Colony size (growth) and redness (viability) provide orthogonal readouts with strong biological signals. These two

*Figure 1 continued on next page*

*Figure 1 continued*

readouts are not correlated ($r_{Pearson}$ = –0.022). Both methods are highly reproducible with overall coefficients of variation of 0.050 and 0.007 for size and redness, respectively (based on 3514 wild-type control colonies across all plates). The lower relative distribution spreads of control values (wild-type vs. entire dataset) indicates a strong biological signal. Fractions of unexplained variance were 0.56 for size and 0.40 for redness.

The online version of this article includes the following figure supplement(s) for figure 1:

**Figure supplement 1.** Overview of conditions used for phenotyping of long intergenic non-coding RNA (lincRNA) knock-out library.

**Figure supplement 2.** Noise, statistical power, and biological signals in phenomics assays.

**Figure supplement 3.** Expression patterns, GC content, and length of long intergenic non-coding RNAs (lincRNAs) studied.

Overall, we collected >1,100,000 phenotype data points for cell growth and >350,000 data points for cell viability. We established a normalization procedure based on control grids to correct for known variations between and within plates which effectively reduces noise in the data (*Figure 1C*, *Figure 1—figure supplement 2A*; *Kamrad et al., 2020b*). Together with the high number of replicates, this normalization provided the statistical power to confidently measure growth differences as small as 5%, thus supporting the detection of subtle lincRNA mutant phenotypes (*Figure 1—figure supplement 2B*). Although control conditions measured in the same batch tended to be more similar, the batch effects remained much smaller than the biological signals (*Figure 1—figure supplement 2C*). Thus, our colony-based phenotyping assays produce robust and reproducible results with high sensitivity. For the lincRNA deletion mutants, we also screened for cell-size and cell-cycle traits using HTP microscopy and flow cytometry analyses (*Figure 1B*). These assays added >20,000 phenotype datasets (microscopic fields analysed), with over 5.7 million cells analysed across 338 samples. Information for all phenotyping conditions is provided in *Supplementary file 1*.

## Phenotyping of deletion mutants in benign conditions

We screened for phenotypes of the lincRNA and coding-gene deletion mutants under benign, standard laboratory conditions using rich and minimal growth media. We looked for mutants showing a significant difference in colony growth and/or colony viability compared to wild-type cells. Among the 141 lincRNA mutants tested, 5 and 10 mutants grew slower than wild-type cells in rich and minimal media, respectively, while 1 mutant grew faster in minimal medium (*Figure 2A*, *Supplementary file 2*). Among the 238 coding-gene mutants tested, 26 and 48 mutants grew slower in rich and minimal media, respectively, while 4 mutants each grew faster in rich and minimal media, 3 of which in both media (*Figure 2A*, *Supplementary files 3 and 4*). Among the total of 51 coding-gene mutants growing slower in our assays, 49 have previously been associated with the phenotype ontology 'abnormal vegetative cell population growth' (*Harris et al., 2013*), thus validating our assay for this phenotype. With respect to colony viability, three lincRNA mutants showed lower viability than wild-type cells, two in rich medium and one in minimal medium (*Figure 2B*, *Supplementary files 2 and 3*). Among the coding-gene mutants, 103 and 42 mutants showed higher or lower viability, respectively, in either or both benign conditions (*Figure 2B*, *Supplementary files 2 and 3*). In conclusion, ~2–7% of the lincRNA mutants showed growth or viability phenotypes compared to ~11–43% of the coding-gene mutants, respectively. These results suggest that coding-gene mutants are more likely to have phenotypes in standard growth conditions. The results also illustrate that colony-viability assays can uncover phenotypes for many additional mutants not evident from colony growth assays (*Kamrad et al., 2020b*; *Lie et al., 2018*).

We examined additional, cellular phenotypes in rich medium for 110 lincRNA deletion mutants. Abnormal cell length or altered duration of cell-cycle stages point to defects in the cell-division cycle. Using HTP microscopy, we determined the length and proportion of binucleated cells; these cells are fully grown and in G1/S phases of the cell cycle. In addition to wild-type cells, we used small *wee1-50* and large *cdc10-129* cell-cycle mutants as controls (*Nurse and Hayles, 2019*). Binucleated wild-type cells showed a median length of 9.7 µm, consistent with published data for ethanol-fixed cells (*Heisler et al., 2014*). Two lincRNA mutants were significantly shorter than wild-type cells and four were longer (*Figure 3A and B*, *Figure 3—figure supplement 1A*, *Supplementary file 2*). Thus, these lincRNAs may be involved in the coordination of cell growth and division. Two of the size mutants, *SPNCRNA.989Δ* and *SPNCRNA.236Δ*, also showed strong slow-growth phenotypes (*Figure 3C*), but no anomalies in cell-cycle phases (*Figure 3D*, *Figure 3—figure supplement 2A and B*). We independently validated

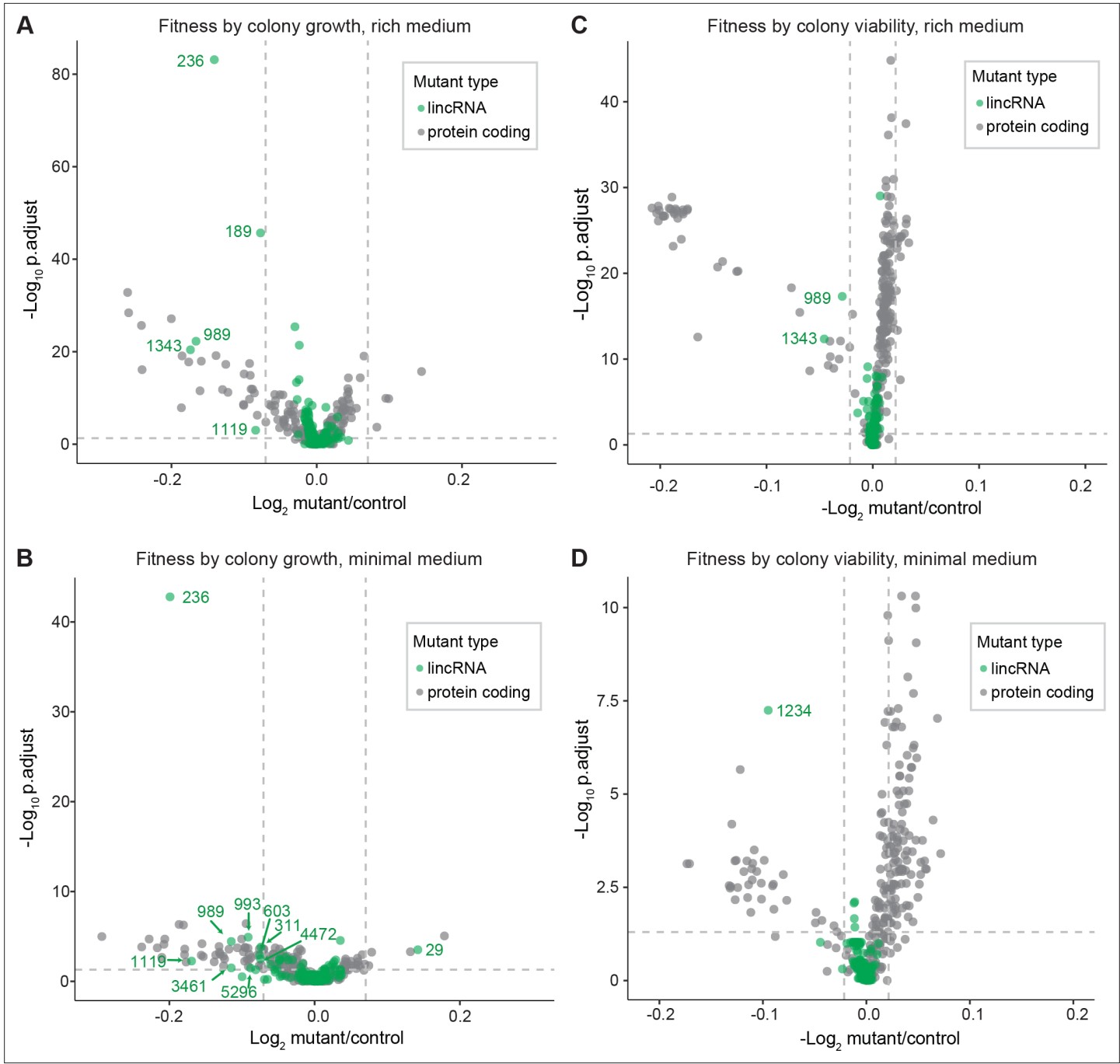

**Figure 2.** Colony growth and viability of deletion mutants in benign conditions. (**A**) Volcano plot for colony size of long intergenic non-coding RNA (lincRNA) mutants (green) and coding-gene mutants (grey) growing in rich medium (top graph) and minimal medium (bottom graph). The dashed lines show the significance thresholds. Strains with lower fitness (smaller colonies) are <0 on the x-axis, and those with higher fitness are >0. We applied a significance threshold of 0.05 after Benjamini–Hochberg correction for multiple testing and a difference in fitness of abs(log2(mutant/wild type))> log2(0.05) to call hits based on colony size; this difference is similar to the median coefficient of variation (CV). (**B**) Volcano plot for colony viability (phloxine B redness score) of lincRNA mutants (green) and coding-gene mutants (grey) growing in rich medium (top graph) and minimal medium (bottom graph). The dashed lines show the significance thresholds. Strains showing lower fitness (redder colonies) are above zero on the x-axis, and those with higher fitness are below zero. We determined quantitative redness scores and applied a significance threshold of 0.05 after Benjamini–Hochberg correction and an effect size threshold of abs(log2(mutant/wild-type))> log2(0.015) to identify colonies that are more or less red than wild-type colonies. The labels indicate the identity of the significant lincRNA genes.

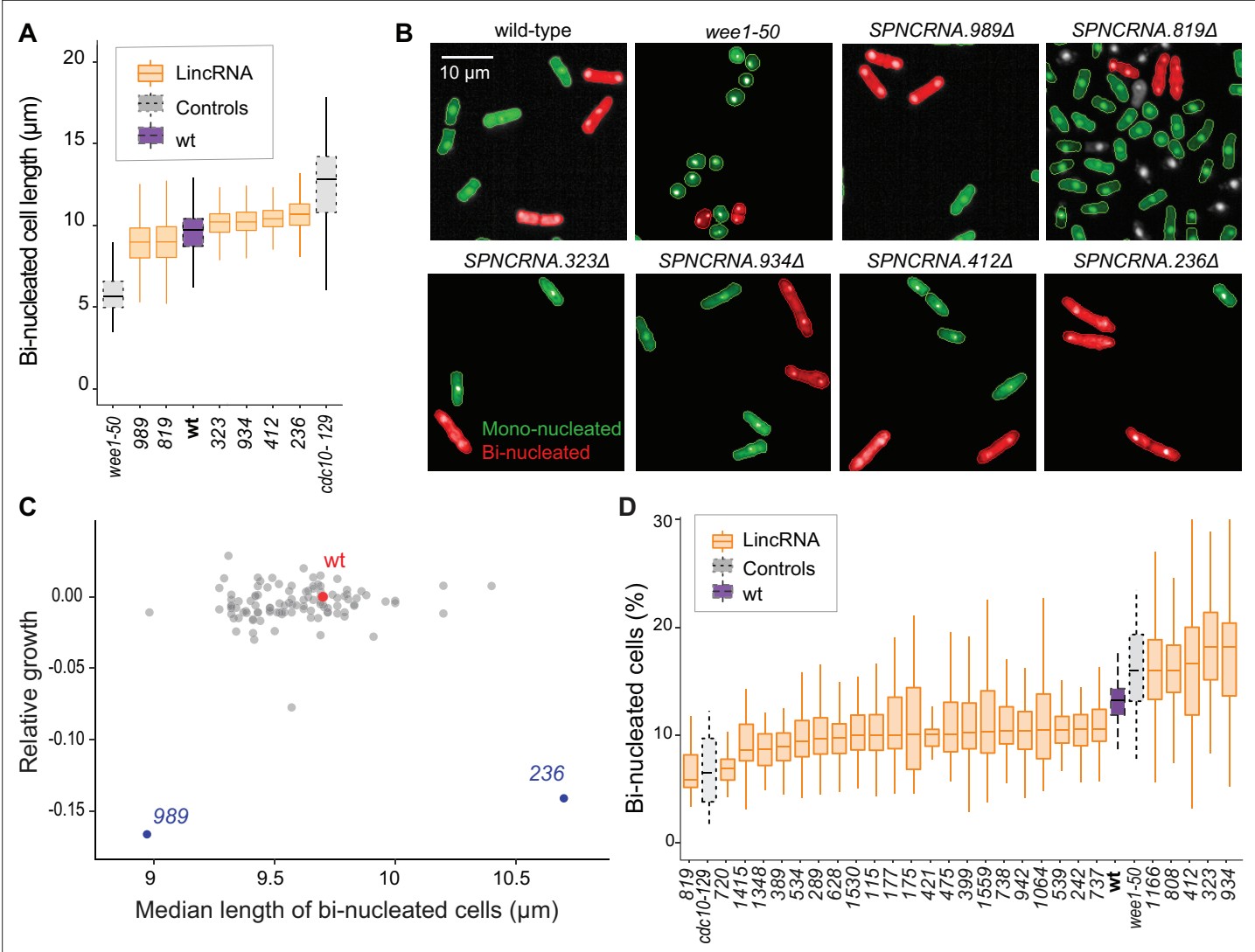

**Figure 3.** Cell-size and cell-cycle traits of long intergenic non-coding RNA (lincRNA) mutants. (**A**) lincRNA deletion mutants showing ≥5% difference in median cell size ($p_{Wilcoxon}<0.05$), compared to wild-type (wt) cells and the conditional cell-size mutants *wee1-50* and *cdc10-129*, captured at 60 min after release to permissive temperature. The sizes of binucleated cells were measured in 63 microscope fields using high-throughput microscopy. (**B**) Representative cells from (**A**), with binucleated cells in red. (**C**) Plot of cell growth vs. cell length of binucleated cells for all lincRNA mutants analysed here. The data on log2 growth of mutant relative to wild-type cells in rich medium are from the colony-based screen (*Figure 2A*). The length data of binucleated cells grown in rich medium are from the high-throughput microscopy (**A**). (**D**) lincRNA deletion mutants showing ≥20% difference in percentage of binucleated cells ($p_{Wilcoxon}<0.05$) compared to wt cells as in (**A**). The median proportion of binucleated cells was quantified from the proportion of binucleated cells in each microscope field, captured for each lincRNA mutant using high-throughput microscopy.

The online version of this article includes the following figure supplement(s) for figure 3:

**Figure supplement 1.** Cell length and binucleated cells for all long intergenic non-coding RNA (lincRNA) mutants.

**Figure supplement 2.** Cell-cycle phenotype analyses using high-throughput flow cytometry and high-throughput microscopy.

the cell-length phenotypes of these two mutants by measuring calcofluor-stained cells growing in rich liquid medium fixed with formaldehyde. This analysis confirmed the shortened average length of *SPNCRNA.989Δ* cells (11.7 ± 0.89 µm; n = 114) and extended median length of *SPNCRNA.236Δ* cells (12.7 ± 0.92 µm; n = 155) compared to wild-type cells (12.1 ± 0.75 µm; n = 129). These two mutants showed a range of other phenotypes and are further discussed below. We also detected phenotypes pointing to defects in transitions between cell-cycle phases: 22 and 5 lincRNA mutants showed significantly reduced and increased proportions of binucleated cells, respectively, compared to the 13.2% binucleated wild-type cells (*Figure 3D*, *Figure 3—figure supplement 1*). Four mutants

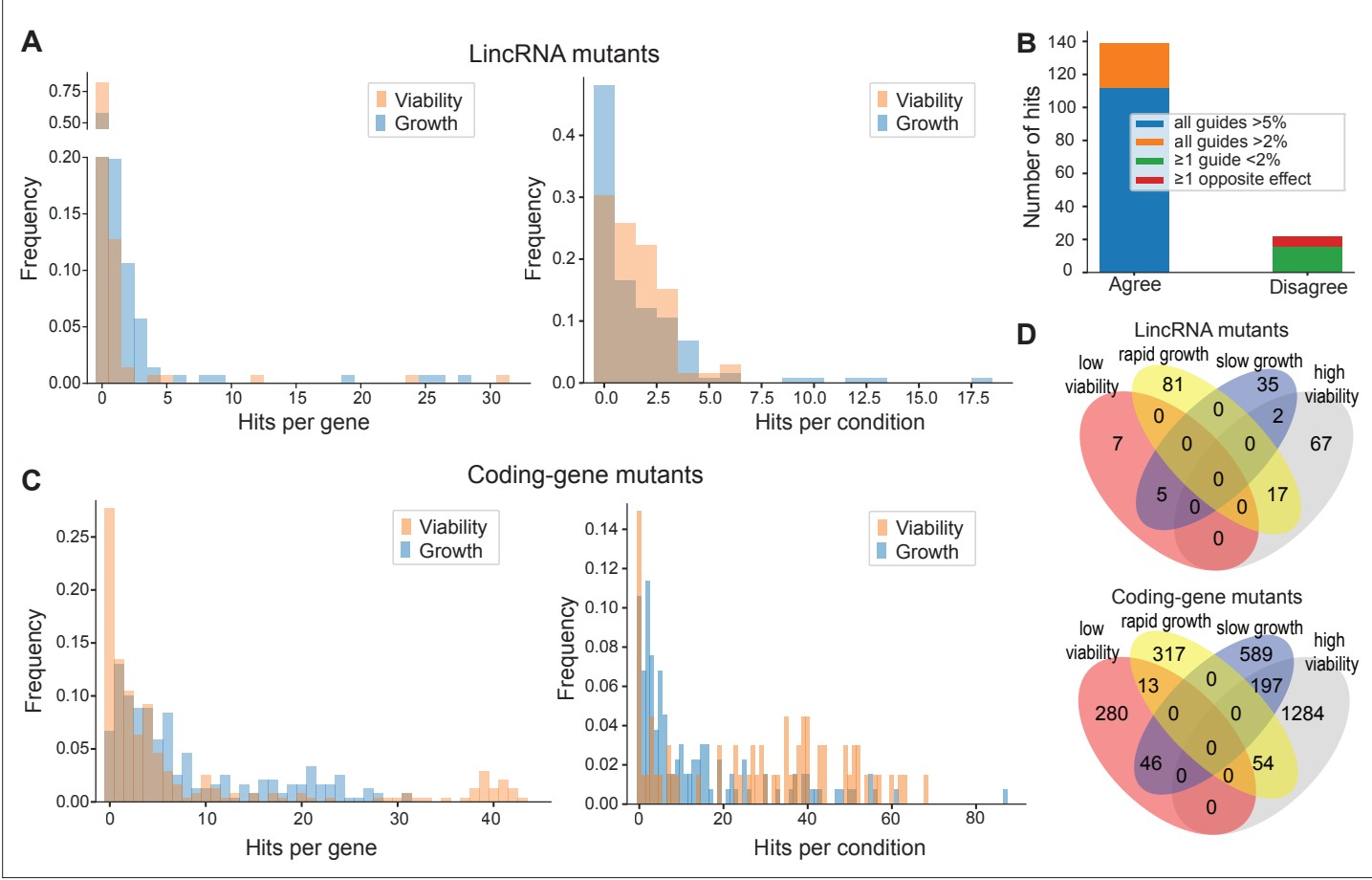

**Figure 4.** Colony growth and viability of deletion mutants in diverse conditions. (**A**) Distributions of significant hits per mutant (left) or per condition (right) for long intergenic non-coding RNA (lincRNA) mutants with altered colony growth (blue) or viability (orange) compared to wild-type cells. (**B**) Plot showing the number of growth phenotype hits agreeing or disagreeing between independently generated lincRNA mutants. (**C**) Distributions of significant hits per mutant (left) or per condition (right) for coding-gene mutants with altered colony growth (blue) or viability (orange) compared to wild-type cells. (**D**) Top Venn diagram: numbers of lincRNA mutants that showed phenotypes for colony growth (rapid or slow) and/or viability (low or high) in 67 conditions. Bottom Venn diagram: numbers of coding-gene mutants showing a phenotype for both colony growth and viability in 67 conditions.

showed both aberrant cell lengths and proportions of binucleated cells: *SPNCRNA.819Δ* cells were shorter and had fewer binucleates, while *SPNCRNA.323Δ*, *SPNCRNA.412Δ,* and *SPNCRNA.934Δ* cells were longer and had more binucleates (*Figure 3A and C*). We validated these microscopy data with HTP flow cytometry, with the results showing a good correlation (*Figure 3—figure supplement 2C*). We conclude that several lincRNAs are involved in regulating cell-size and/or cell-cycle progression.

## Phenotyping of deletion mutants in multiple nutrient, drug, and stress conditions

We assayed for colony size (growth) phenotypes of the lincRNA and coding-gene mutants in the presence of various stresses or other treatments, relative to the same mutants growing in benign conditions and normalized for wild-type growth (Materials and methods). To this end, we applied the same significance thresholds as for benign conditions. Among the 141 lincRNA mutants tested, 60 (43%) showed growth phenotypes in at least one condition (*Supplementary files 2 and 3*). Together, these 60 mutants showed 211 growth phenotypes across conditions, with 69 of the 145 conditions producing phenotypes in at least one mutant (*Figure 4A*). The 211 hits included 150 resistant and 61 sensitive phenotypes (i.e. mutants showing larger or smaller colonies, respectively, in assay conditions compared to the control condition, each relative to wild-type). Seven lincRNA mutants showed growth phenotypes in at least five conditions, with *SPNCRNA.236Δ* showing the most phenotypes, being resistant in 26 and sensitive in two conditions (*Supplementary files 2 and 3*). Among all conditions,

the most phenotypes were triggered by 0.075% MMS (causing DNA damage; 13 hits) and Brefeldin A (inhibiting protein transport from endoplasmic reticulum to Golgi; 18 hits).

Due to possible off-target mutations introduced by CRISPR/Cas9, we generated independent deletion mutants using different guide RNAs targeting the same lincRNA gene. These independent mutants generally produced highly similar growth phenotypes (*Figure 4B*). Of the 161 phenotypes associated with lincRNAs represented by two or more independent mutants, 112 phenotypes agreed between the corresponding mutants (all mutants showed median effect sizes [MES] > 5%), and 27 hits showed a similar trend (MES > 2%). In 16 cases, at least one guide RNA showed no phenotype (MES < 2%), and in only 6 cases did the guide RNAs show opposite effects (*Figure 4B*). These results indicate that any secondary effects from CRISPR/Cas9-based gene deletions did not affect the consistency of our phenotype results in the vast majority of cases.

Among the 238 coding-gene mutants tested, 223 (93.7%) showed growth phenotypes in at least one condition, 104 of which represent priority unstudied genes that have remained entirely uncharacterized (*Wood et al., 2019*). Together, these 223 mutants showed 1924 growth phenotypes across conditions, with 119 of the 145 conditions tested producing phenotypes in at least one mutant (*Figure 4C*, *Supplementary files 3 and 4*). The 1924 hits included 651 resistant and 1273 sensitive phenotypes.

We also assayed for colony-viability phenotypes of the lincRNA and coding-gene mutants across stress or other treatments relative to mutant cells growing in benign control conditions, normalized for wild-type growth. To this end, we applied the same quantitative redness scores and significance thresholds as for the benign conditions. Among the 141 lincRNA mutants tested, 25 (17.7%) differed in viability in at least one condition compared to wild-type cells (*Supplementary file 2*). Together, these 25 mutants showed 98 phenotype hits across conditions, with 45 of the 67 conditions tested producing phenotypes in at least one mutant (*Figure 4A*). The 98 hits included 86 resistant and 12 sensitive phenotypes (higher and lower viability than wild-type, respectively). Two lincRNA mutants, which were sensitive in the benign condition, caused ~56% of the hits, all resistant, in conditions that partially suppressed this sensitive phenotype: *SPNCRNA.989Δ* (31 hits) and *SPNCRNA.1343Δ* (24 hits) (*Supplementary file 2*). These lincRNAs are discussed further down. Among all conditions, the highest number of hits with viability phenotypes were observed in rich medium with 0.5 M KCl or with 0.005% MMS (six hits each) and in minimal medium with canavanine (five hits).

Among the 238 coding-gene mutants tested, 172 (72.3%) showed viability phenotypes in at least one condition. Together, these 172 mutants showed 1874 phenotype hits across conditions, with 57 of the 67 conditions tested producing phenotypes in at least one mutant (*Figure 4C*, *Supplementary files 3 and 4*). The 1874 hits included 1535 resistant and 339 sensitive phenotypes.

We then explored the relationships between colony growth and viability for the 67 conditions used to measure both phenotypes. The lincRNA mutants produced 140 growth phenotypes and 98 viability phenotypes, but in only 24 instances were both phenotypes associated with the same mutant (*Figure 4D*). The coding-gene mutants showed 1216 growth phenotypes and 1874 viability phenotypes, with only 310 instances where both phenotypes were associated with the same mutant (*Figure 4D*). A large excess of high-viability phenotypes was evident for coding-gene and, even more so, for lincRNA mutants (*Figure 4D*). Thus, slowly growing mutants did often show higher viability rather than lower viability, especially in coding-gene mutants. Together, these results further highlight that the colony-viability assays produce orthogonal phenotype information to the colony growth assays and can uncover many additional phenotypes (*Kamrad et al., 2020b*; *Lie et al., 2018*).

The lincRNAs that showed phenotypes were distributed across the genome (*Figure 1A*). They were not enriched in any particular gene expression patterns, showing diverse responses to genetic or physiological perturbations (*Figure 1—figure supplement 3A*). The lincRNAs associated with phenotypes were of similar length as those without phenotypes, but they tended to have a higher GC content (*Figure 1—figure supplement 3B*). This result raises the possibility that the GC content reflects or even determines the likelihood of lncRNA function.

In conclusion, substantial proportions of the lincRNA mutants showed growth (43%) and/or viability (18%) phenotypes in some stress conditions, and the majority of coding-gene mutants showed phenotypes in these conditions (72–94%). With respect to viability phenotypes, much larger proportions of both lincRNA and coding-gene mutants were resistant (87.8% and 81.9%, respectively). This bias could partly reflect that many mutants are growing somewhat more slowly in benign conditions

(*Figure 2A*), a trade-off that may render them more resilient to stresses (*López-Maury et al., 2008*). Together, these analyses show that phenomics assays can effectively uncover functional clues not only for protein-coding genes but also for many lincRNAs.

## Integrated analyses of functional signatures from deletion mutants

Using unsupervised clustering, we mined the rich deletion-mutant phenotype data to explore functional profiles for both protein-coding and lincRNA genes. For the phenotype calling described above, we wanted to identify functional clues and gene-environment interactions with high confidence (i.e. low false discovery rate). Here, using a less conservative analysis, we applied a multivariate, global approach by converting effect sizes to a modified z-score to indicate the deviation from the expected phenotype value in units of standard deviations from the wild-type control in the same condition. Several conditions involved the same stressor, for example, the same drug used at different doses (*Supplementary file 1*). We aggregated such related conditions and used the strongest median response for each mutant and set of conditions (Materials and methods). The protein-coding mutants generally showed stronger phenotypes than the lincRNA mutants as measured by the magnitude of the effect sizes (*Figure 5—figure supplement 1A*). To compare phenotypes across the two types of mutants, we discretized the data, classing mutants as either sensitive (–1), resistant ( + 1), or similar (0) to their fitness in the corresponding control condition (*Figure 5—figure supplement 1B*, *Supplementary file 5*). Thresholds were chosen at ±1.5 standard deviations for both growth and viability data, which resulted in ~23% of all data points classed as non-zero in each dataset. We limited this analysis to 41 sets of aggregated 'core' conditions in which all mutants were phenotyped (*Supplementary file 5*).

Applying this analysis, most lincRNA mutants showed few or no phenotypes across the 41 core conditions, while 16 lincRNA mutants showed strong phenotype profiles across many conditions. Such uneven distribution in the phenotype numbers associated with lincRNAs indicates that the data reflect biology rather than technical noise. In total, 194 mutants showed a phenotype in five or more sets of conditions, including the 16 lincRNA mutants, and these mutants were used for hierarchical clustering. Clear patterns were evident, and we divided the genes into three main clusters (*Figure 5A*, *Figure 5—figure supplement 1C*, *Supplementary file 5*). Clusters 1, 2, and 3 contained 2, 10, and 4 lincRNAs, respectively, providing an opportunity to infer function through 'guilt by association' with known protein-coding genes in the same clusters. This approach was somewhat limited because only 115 of the 178 protein-coding genes in the clusters had known or inferred biological roles. Using the AnGeLi tool (*Bitton et al., 2015*), we identified functional enrichments for the clusters as described below.

Cluster 1 showed the most defined phenotype signature, characterized by many mutants displaying higher viability in 15 stress conditions, lower viability in the benign conditions and in canavanine A, and slow growth in benign conditions and several drugs tested, including hydrogen peroxide and antimycin A (*Figure 5A*, *Figure 5—figure supplement 1C*, *Supplementary file 5*). This cluster was enriched in various GO categories related to protein localization/transport, cellular respiration, phosphate metabolism, and protein translation (the latter including five cytosolic/mitochondrial ribosomal subunits, six translation factors, and three subunits of the elongator complex). The cluster also included nine genes involved in nutrient- or stress-dependent signalling (*Supplementary file 5*). With respect to phenotype ontology (*Harris et al., 2013*), this cluster was enriched in multiple terms related to cytoskeleton aberrations, abnormal respiration and translation, as well as altered cell growth and stress sensitivity. Indeed, 80% of the mutants in this cluster have previously been associated with decreased cell population growth, and 87% are associated with increased sensitivity to chemicals. These enrichments validate our phenotype data.

Cluster 1 contained the two lincRNAs, *SPNCRNA.989* and *SPNCRNA.1343*, which accounted for ~56% of the colony-viability phenotypes among the lincRNA deletion mutants. When overexpressed, however, they generated just 1–2 hits, much fewer than average (see below). This pattern suggests that these lincRNAs may function in *cis*, regulating nearby genes. Notably, both lincRNAs are located upstream of genes regulated by the Pho7 transcription factor (*Schwer et al., 2017*), which functions during phosphate starvation and other stresses (*Carter-O'Connell et al., 2012*): *SPNCRNA.989* and *SPNCRNA.1343* are divergently expressed to *atd1* and *tgp1*, respectively (*Figure 5—figure supplement 2*). *SPNCRNA.1343* partially overlaps with the *nc-tgp1* RNA that regulates phosphate

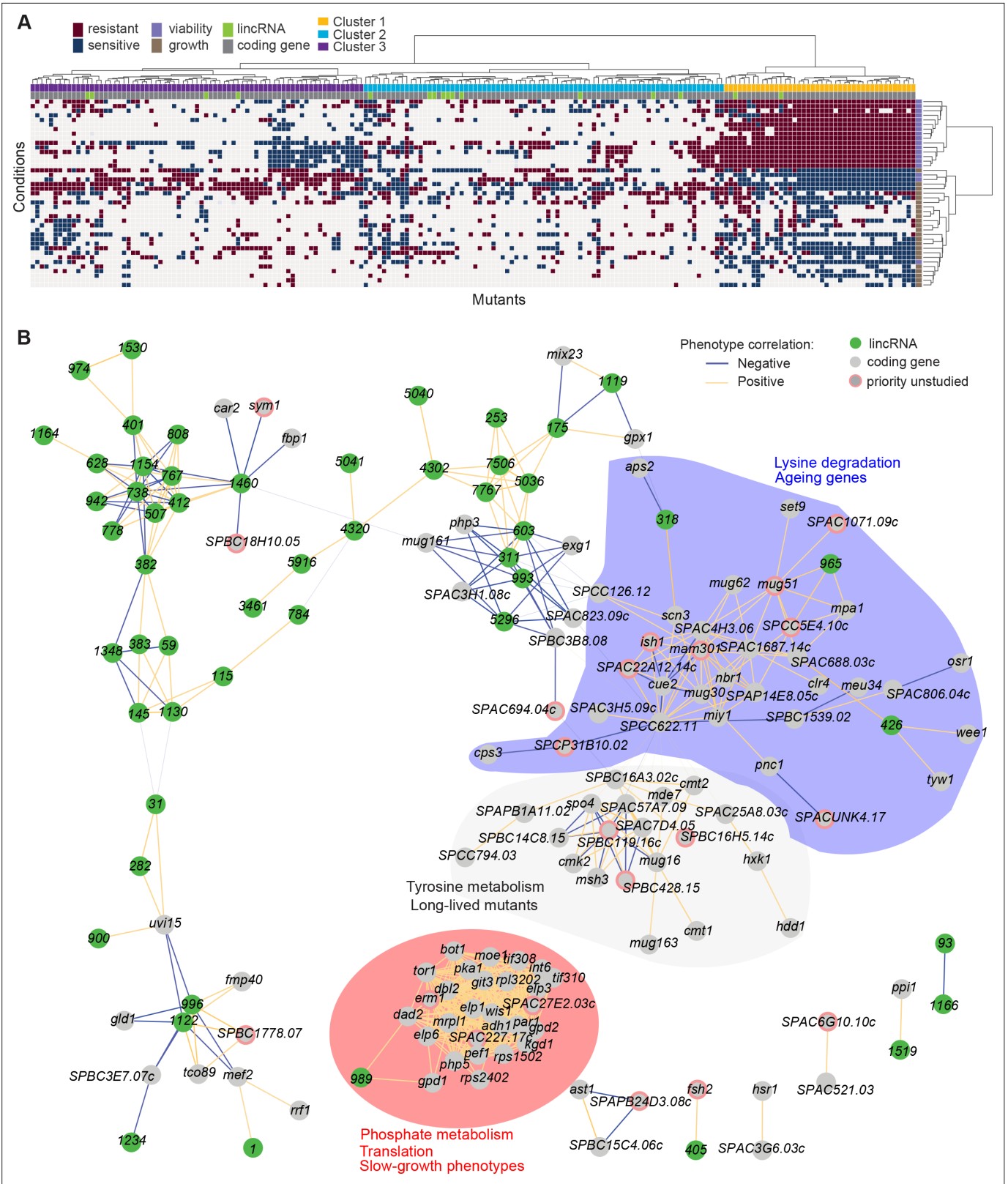

**Figure 5.** Functional signatures in long intergenic non-coding RNA (lincRNA) phenotype profiles. (**A**) Hierarchical clustering of discretized data for 16 lincRNA mutants (green) and 178 coding-gene mutants (grey), as indicated above the columns. Clustering was limited to the core conditions where phenotypes for all mutants were available, including growth phenotypes (brown) and viability phenotypes (purple), as indicated to the right of rows. Only mutants with at least five hits across the 41 conditions are shown. Resistant (dark red) and sensitive (dark blue) phenotypes are indicated for

*Figure 5 continued on next page*

*Figure 5 continued*

corresponding mutant-condition combinations. Hierarchical clustering of both mutants and conditions was performed with the Ward method using Euclidean distances. Based on the dendrogram, the genes were divided into three clusters indicated in different colours (top row). A detailed version of this cluster specifying the conditions and mutants is provided in *Figure 5—figure supplement 1C*. (**B**) Cytoscape gene network representing phenotype correlations between lincRNA and coding-gene mutants. Yellow and blue edges show positive and negative phenotype correlations, respectively. The lincRNAs are shown in green and the protein-coding genes in grey, including a pink border if their function is unknown. Clusters discussed in the main text are highlighted in colour.

The online version of this article includes the following figure supplement(s) for figure 5:

**Figure supplement 1.** Effect sizes, data discretization, and cluster details.

**Figure supplement 2.** Genome browser view of the chromosomal regions surrounding *SPNCRNA.1343* (left) and *SPNCRNA.989* (right).

**Figure supplement 3.** Spot assays with fivefold serial dilutions to validate selected long intergenic non-coding RNA (lincRNA) deletion phenotypes from the screen.

homeostasis by repressing the adjacent *tgp1* gene via transcriptional interference; deletion of *SPNCRNA.1343* has been shown to increase *tgp1* expression by inhibiting *nc-tgp1* expression (*Ard et al., 2014*; *Ard and Allshire, 2016*; *Garg et al., 2018*; *Shah et al., 2014*; *Yague-Sanz et al., 2020*). Inspection of the region upstream of *SPNCRNA.989* suggested a regulatory mechanism similar to *tgp1*, with divergent transcripts towards *atd1* likely driven by a bidirectional promoter from the nucleosome-depleted region upstream of *SPNCRNA.989* (*Figure 5—figure supplement 2*). These patterns suggest that additional Pho7-regulated genes, like *atd1*, are controlled via upstream RNAs, similar to the *tgp1, pho84,* and *pho1* genes that respond to phosphate limitation (*Carter-O'Connell et al., 2012*). The similar phenotypes of *SPNCRNA.989Δ* and *SPNCRNA.1343Δ* mutants therefore suggest that these lincRNA deletions interfere with the expression of their neighbouring genes and thus with processes affected by this regulon. In spotting assays, the phenotypes of *SPNCRNA.989Δ* and *SPNCRNA.1343Δ* often differed from those of *tgp1Δ* and *atd1Δ* (*Figure 5—figure supplement 3*). These results are consistent with the lincRNA deletion leading to induction, rather than repression, of their coding-gene neighbours.

Cluster 2 contained a majority of genes that are not associated with any functional annotations, including 10 lincRNAs genes (*Figure 5—figure supplement 1C*, *Supplementary file 5*). This cluster was enriched for long-lived mutants and for genetic interactions (based on Biogrid data; *Breitkreutz et al., 2007*), meaning that the protein-coding genes within this cluster are approximately four times more likely to interact with each other than expected by chance. This cluster included seven genes involved in stress and/or nutrient signalling pathways and six genes for transcription factors functioning during stress/nutrient responses or in unknown processes. The phenotype data in this cluster were sparse and lacked a convincing functional signature across the coding and lincRNA genes.

Cluster 3 was characterized by most mutants showing rapid growth in VPA, formamide, and sodium orthovanadate, and many of these mutants also showed higher viability in benign conditions but lower viability in VPA. This cluster was enriched for long-lived mutants and for energy metabolism, including four genes each functioning in glycolysis and the TCA cycle. Intriguingly, one of the four lincRNA genes in this cluster, *SPNCRNA.236*, is located upstream the pyruvate-kinase gene *pyk1*, which is involved in the last step of glycolysis to generate pyruvate for the TCA cycle or fermentation. The finding that the *SPNCRNA.236Δ* mutant leads to a similar phenotypic signature as does deletion of glycolysis or TCA cycle genes raises the possibility that *SPNCRNA.236* acts in *cis* to control *pyk1* expression. Consistent with this idea, *SPNCRNA.236Δ* mutants grow slowly (*Figure 2A*) while increased activity of Pyk1 leads to faster growth (*Kamrad et al., 2020a*). However, *SPNCRNA.236* also generates phenotypes in 11 conditions when overexpressed from a plasmid (*Supplementary file 6*), including faster growth in minimal medium, which is the opposite of the slower growth of the deletion mutant in the same condition. Thus, it is also possible that *SPNCRNA.236* can act in *trans*.

We validated phenotypes of five lincRNA deletions from clusters 1 and 3 as well as deletions from neighbouring coding genes using serial dilution spotting assays under 13 conditions. Detection of subtle phenotypes involving 5% differences in growth is difficult with such spotting assays. Nevertheless, we could confirm 11 (84%) of the phenotypes detected by the HTP colony-based assays (*Figure 5—figure supplement 3*). We conclude that there is generally a good agreement between these different phenotyping assays.

To further explore our dataset, we discretized all deletion-mutant phenotypes, both from lincRNAs and coding genes (*Supplementary files 2-4*). Pearson correlations of phenotype profiles were then used for constructing a network that was visualized with Cytoscape (*Shannon et al., 2003*). The network included several distinct clusters. A large, tight cluster consisted mostly of protein-coding genes (*Figure 5B*, highlighted in red). This cluster, which was similar to cluster 1 (*Figure 5A*), included *SPNCRNA.989* and was enriched for genes involved in phosphate metabolism and translation, and 89% of the mutants in this cluster displayed slow growth phenotypes (*Figure 5B*). Another large cluster was enriched for lysine metabolism with 18% of the mutants showing ageing-related phenotypes such as increased lifespan during quiescence (*Sideri et al., 2014*). This cluster also included three lincRNAs: *SPNCRNA.318, SPNCRNA.426,* and *SPNCRNA.965* (*Figure 5B*, highlighted in blue). Network analysis of the whole phenotypic dataset revealed further connections between several lincRNAs and coding genes. For example, a negative phenotypic correlation was evident between *SPNCRNA.1460* and *fbp1* (*Figure 5B*, upper left); *fbp1* is a key gene responding to glucose starvation that is regulated by upstream non-coding RNAs (*Hirota et al., 2008*; *Hoffman and Winston, 1990 Oda et al., 2015*). *SPNCRNA.1460* is located upstream of *scr1*, encoding a transcriptional repressor that negatively regulates *fbp1* (*Tanaka et al., 1998*; *Vassiliadis et al., 2019*). This link raises the possibility that *SPNCRNA.1460* controls *scr1* expression and, therefore, *fbp1* expression. The same cluster also included *car2*, which is also implicated in carbon metabolism, and two priority unstudied genes, whose association suggests that they function in similar processes. Interestingly, some clusters consisted exclusively or mostly of lincRNAs (*Figure 5B*, upper left). Naturally, these clusters showed no functional enrichments, but they point to several lincRNAs acting in related cellular processes, possibly together.

## Phenotyping of lincRNA overexpression mutants in multiple conditions

Gene overexpression provides complementary phenotype information to gene deletion (*Prelich, 2012*). We constructed strains that ectopically overexpressed 113 lincRNAs from a plasmid under the strong *nmt1* promoter in minimal medium (Materials and methods). A real-time quantitative PCR (RT-qPCR) analysis of eight overexpression constructs showed that the lincRNAs were 35- to 2200-fold overexpressed relative to the empty-vector control strain, which expresses the lincRNAs at native levels (*Figure 6—figure supplement 1A*).

We then looked for differences in colony growth under benign conditions compared to empty-vector control cells. We also looked for growth phenotypes in the presence of various stresses or other treatments, relative to growth in benign control conditions and normalized for the growth of empty-vector control cells. In the benign condition, most lincRNA overexpression strains grew faster compared to the empty-vector control. This pattern may reflect an indirect effect of lincRNA transcription by increasing plasmid copy numbers and/or expression of the budding yeast *LEU2* marker that is limiting for growth. Therefore, we normalized the colony growth of overexpression mutants in the stress conditions by the growth in the benign condition to correct for this potential bias. We used a more stringent significance threshold for the overexpression mutants than for the deletion mutants (*Figure 6A*) because ectopic overexpression of genes involves cell-to-cell variation in plasmid copy numbers, leading to higher phenotypic heterogeneity (*Siam et al., 2004*). Among the 113 lincRNA overexpression strains tested, 102 (90.3%) showed growth phenotypes in at least one condition (*Figure 6A*, *Supplementary file 6*). Together, these 102 overexpression strains showed 565 growth phenotypes across conditions. The 565 hits included 347 resistant and 218 sensitive phenotypes (i.e. mutants showing larger or smaller colonies, respectively, in the assay condition than in the control). 14 lincRNA overexpression strains showed more consistent phenotypes in 10 or more conditions, topped by *SPNCRNA.335* that showed sensitive and resistant phenotypes in 12 and 3 conditions, respectively (*Figure 6—figure supplement 1B*). No clear pattern was evident between expression levels and phenotype hits, for example, lincRNAs without phenotypes when overexpressed showed similar fold-changes as a lincRNA showing 13 phenotypes (*Figure 6—figure supplement 1A*).

With respect to the 47 conditions tested, 42 produced phenotypes in at least one mutant (*Supplementary file 6*). Over 80% of the 565 phenotypes came from only 21 of the 47 conditions, and ~24% of the phenotypes came from just three conditions: proline as a nitrogen source, 5 mM valproic acid (VPA), and 10 mM hydroxyurea (HU) (*Figure 6—figure supplement 1C*). Proline is a poor nitrogen source which causes nitrogen stress and slows growth (*Davie et al., 2015*). Notably, the expression of

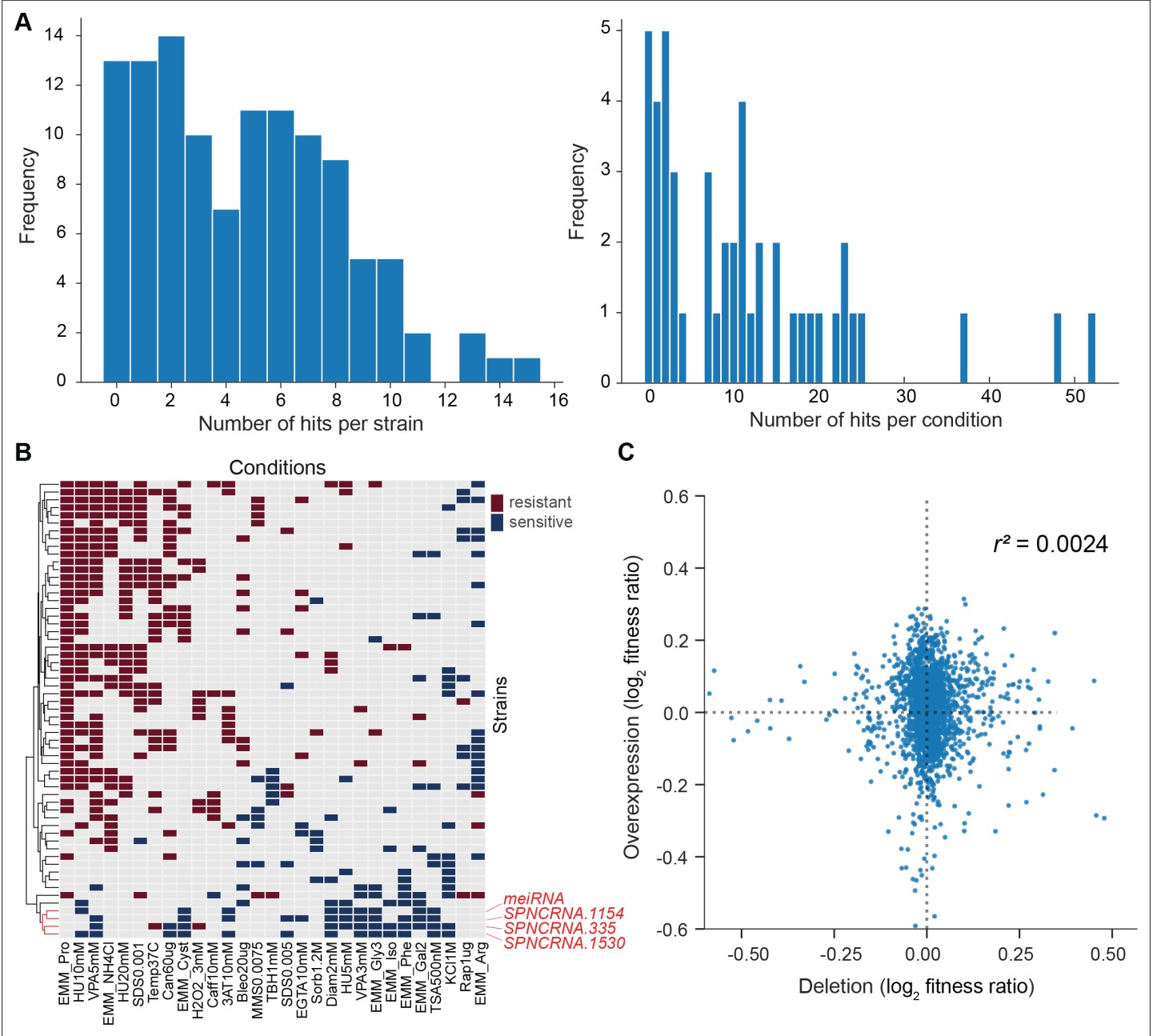

**Figure 6.** Growth phenotypes of long intergenic non-coding RNA (lincRNA) overexpression mutants in different conditions. (**A**) Distributions of significant phenotype hits per strain (left) and condition (right) for lincRNA overexpression strains with altered growth under benign and stress conditions. Overall, 113 overexpression strains were phenotyped under 47 different conditions, based on 31 distinct environmental factors. We applied a significance threshold of p≤0.01, after correction for multiple testing, and a difference in fitness of ≥5% to call hits based on colony size. (**B**) Hierarchical clustering of discretized relative log2 median effect sizes for lincRNA overexpression strains (rows) using only the strains and conditions with at least five hits (59 mutants, 29 conditions). Resistant (red) and sensitive (blue) phenotypes are indicated for strain-condition combinations. The sub-cluster highlighted in red is discussed in the main text. (**C**) Comparison of phenotype data from lincRNA deletion vs. overexpression mutants. Plot showing maximum median effect sizes for 104 lincRNA mutants represented in both deletion and overexpression libraries, phenotyped under 22 shared conditions. The pairwise Pearson correlation coefficient is indicated. To aid visualization, 10 extreme outliers were removed out of 2288 data points.

The online version of this article includes the following figure supplement(s) for figure 6:

**Figure supplement 1.** Expression fold-changes and phenotypes per strain or condition for long intergenic non-coding RNA (lincRNA) overexpression strains.

**Figure supplement 2.** Meiotic phenotypes of long intergenic non-coding RNA (lincRNAs) from sub-cluster in *Figure 6B*.

many lincRNAs is strongly induced under nitrogen starvation (*Atkinson et al., 2018*). Together with the results presented here, this indicates that several lincRNAs function during nitrogen stress. VPA is an inhibitor of histone deacetylases, and some lincRNAs are involved in histone modification (*Rinn and Chang, 2012*). Roles for lincRNAs in neurological disorders such as epilepsy are also emerging, and VPA can ameliorate epilepsy partly by repressing some of these lincRNAs (*Hauser et al., 2018*). HU inhibits DNA synthesis, and our findings suggest that several lincRNAs are involved in related processes. For example, some lincRNAs have been implicated in double-strand break repair (*Bader et al., 2020*). We propose that several lincRNAs can influence cellular growth in *trans* when ectopically overexpressed under conditions that affect certain cellular processes.

We looked for functional signatures in the lincRNA overexpression phenotypes using hierarchical clustering (*Figure 6B*). Unlike for lincRNA deletion mutants, this dataset did not provide the functional context from coding-gene mutants. We observed a conspicuous sub-cluster of four lincRNA overexpression strains that showed slower growth in many of the conditions (*Figure 6B*, highlighted in red). One of these four overexpressed lincRNAs was the well-characterized *meiRNA* that functions in the induction of meiosis (*Watanabe and Yamamoto, 1994*). In mitotically growing cells, *meiRNA* binds to Mmi1 via its DSR motif and is degraded by the nuclear exosome, while upon induction of meiosis, *meiRNA* binds to the RRM motif of Mei2, which in turn promotes meiosis (*Yamashita, 2019*). The other three lincRNAs in this sub-cluster contain motifs for potential Mei2 binding, including two DSR motifs. Moreover, like *meiRNA,* the other three lincRNAs are also de-repressed in nuclear-exosome mutants and during meiosis (*Atkinson et al., 2018*). Together, these findings raise the possibility that the three unknown lincRNAs in the sub-cluster also function in meiosis. To test this hypothesis, we deleted these three lincRNA genes, along with the meiRNA gene, in a homothallic $h^{90}$ background (to allow self-mating). We then analysed meiosis and spore viability of these four deletion strains together with a wild-type control strain. While cell mating was normal in the deletion mutants (*Figure 6—figure supplement 2A*), meiotic progression was somewhat delayed in *SPNCRNA.1154Δ* and *SPNCRNA.1530Δ* mutants as well as, most strongly, in *meiRNAΔ* mutants (*Figure 6—figure supplement 2B*), with the latter reported before (*Yamashita, 2019*). Notably, all four deletion mutants showed significantly reduced spore viability compared to the control strain (*Figure 6—figure supplement 2C*). As predicted by the clustering analysis, these results indicate that *SPNCRNA.1154, SPNCRNA.1530,* and *SPNCRNA.335* play roles in meiotic differentiation.

We compared the phenotype data from deletion and overexpression mutants. We obtained data for both types of mutants for 104 lincRNAs in 22 conditions, 18 of which showed phenotypes. Of these 104 lincRNAs, only 7 did not produce any phenotypes in any condition tested. Under the 18 conditions, a higher proportion of lincRNA overexpression mutants (86.5%) than deletion mutants (32.7%) produced phenotype hits in at least one condition. Moreover, lincRNA overexpression generally resulted in larger effect sizes, that is, stronger phenotypes, than did lincRNA deletion (*Figure 6C*). Similar trends have been reported for coding-gene mutants. For example, 646 and 1302 growth phenotypes are caused by deletion and overexpression mutants, respectively, of non-essential budding yeast genes (*Yoshikawa et al., 2011*), and 64 transcription factor genes of fission yeast show growth phenotypes when overexpressed but not when deleted (*Vachon et al., 2013*). Only a few lincRNAs showed phenotypes as both deletion and overexpression mutants. For example, *SPNCRNA.236* showed rapid growth (resistant) phenotypes in both overexpression and deletion mutants in five conditions, while in the benign condition, the overexpression and deletion mutant showed rapid and slow growth, respectively (*Supplementary files 2 and 6*). In general, our phenotype data for lincRNA deletion and overexpression mutants showed little overlap and poor correlation (*Figure 6C*). These results illustrate the complementary information provided by these two types of mutants.

How might lincRNA overexpression result in more phenotypes than lincRNA deletion? Overexpression of a protein-coding gene can burden cells via resource-consuming translation or toxic protein levels (*Bolognesi and Lehner, 2018*; *Moriya, 2015*). Although overexpression of a single lincRNA should not affect resource allocation, it is possible that these AT-rich RNAs engage in non-specific molecular interactions. Recent findings indicate that RNAs are assembly prone and must be tightly regulated as they can promote paraspeckles, stress granules, and phase separation (*Fox et al., 2018*; *Van Treeck et al., 2018*). Such processes could trigger the overexpression phenotypes in certain physiological conditions, reflecting that lincRNAs are biologically active molecules. Nevertheless, the observed 565 phenotype hits amount to only 10.6% of the potential 5311 hits if overexpression of

all 113 lincRNAs caused phenotypes in the 47 conditions tested (*Supplementary file 6*). Thus, over-expressed lincRNAs do not generally lead to any non-specific or toxic effects, and the observed phenotypes may, therefore, mostly reflect specific lincRNA functions. Given that many lincRNAs may function in specialized conditions (*Atkinson et al., 2018*; *Cabili et al., 2011*; *Derrien et al., 2012*; *Hon et al., 2017*; *Pauli et al., 2012*), deletion mutants will only reveal phenotypes when assayed in the relevant conditions. On the other hand, the 'gain-of-function' overexpression mutants may also reveal phenotypes in conditions where the lincRNAs do not normally function. Notably, phenotypes that are caused by lincRNAs being ectopically expressed from plasmids point to a function that is exerted in *trans*, via the lincRNA itself, rather than via its transcription or other *cis* effects. Our findings therefore raise the possibility that many of the lincRNAs tested can function over a distance.

## Conclusions

We applied a phenomics approach to explore the functional importance of *S. pombe* lincRNAs, including colony-based and cellular assays of deletion mutants and colony-based assays of overexpression strains. A panel of deletion mutants of coding genes were screened in parallel for comparison and functional context. Together, these assays revealed phenotypes for 84 of 141 deleted lincRNAs, 229 of 238 deleted coding genes, and 102 of 113 overexpressed lincRNAs. This extensive phenotyping uncovers lincRNAs that contribute to cellular resistance or sensitivity in specific conditions, reflected by altered colony growth and/or viability, and lincRNAs that are involved in the size control and the cell-division cycle. Systematic screening for genetic interactions between lincRNA and coding-gene mutants (*Dixon et al., 2008*) could provide valuable clues about functional relationships. As expected, higher proportions of coding-gene mutants showed phenotypes, and these phenotypes tended to be stronger (larger effect sizes) than for lincRNA mutants. In benign conditions, the lincRNA mutants were ~3- and 30-fold less likely to show phenotypes for colony growth or viability, respectively, than coding-gene mutants. This difference was less pronounced in the nutrient, drug, and stress conditions, where many more lincRNA mutants revealed phenotypes, at only approximately two- to fourfold lower proportions than coding-gene mutants for colony growth or viability, respectively. Moreover, compared to lincRNA deletion mutants, the lincRNA overexpression strains were approximately twofold more likely to show phenotypes, which also tended to be stronger. As predicted from clustering of overexpression phenotypes, deletion mutants for three lincRNAs showed defects in meiotic differentiation. Together, these findings support the notion that most lincRNAs play specialized roles in specific conditions. The findings also indicate that lincRNAs in general have subtler functions than proteins, for example, in fine-tuning of gene expression. Accordingly, it was important that our HTP assays were highly sensitive to detect subtle phenotypes. We conclude that a substantial proportion of lincRNAs exert cellular functions under certain conditions, and many of which may act in *trans* as RNAs. This analysis provides a rich framework to mechanistically dissect the functions of these lincRNAs in the physiologically relevant conditions.

# Materials and methods
## Deletion and overexpression strain libraries

Using a CRISPR/Cas9-based approach and primer design tool for seamless genome editing (*Rodríguez-López et al., 2016*), we deleted 141 different lincRNA genes located across all the *S. pombe* chromosomes (*Figure 1A*; see *Supplementary file 1* for coordinates). In total, 113 lincRNA genes were deleted in the *972 h⁻* background, and 73 lincRNA genes were deleted in the *968 h⁹⁰* background, the latter including 15 newly identified lincRNAs (*Atkinson et al., 2018*). 30 lincRNAs were deleted with one guideRNA (gRNA), 103 were deleted using two gRNAs, and 8 were deleted using three gRNAs. All lincRNA deletion strains were checked for missing open-reading frames by PCR, and for 20 strains we also sequenced across the deletion scars (*Rodríguez-López et al., 2016*). We rechecked all strains by PCR after arraying them onto the 384 plates to ensure that no errors occurred during the process. For the protein-coding deletion mutants, we generated a prototroph version of Bioneer V.5 deletion library (*Kim et al., 2010*) as described (*Malecki and Bähler, 2016*). Strains were arranged into 384-colony format using a RoToR HDA colony-pinning robot (Singer Instruments), including a 96-colony grid of wild-type *972 h⁻* strains for plate normalization (*Kamrad et al., 2020b*). We selected a subset

of genes to broadly cover all main GO categories, together with 91 uncharacterized genes. *Supplementary file 1* provides information on the individual strains.

We generated ectopic overexpression constructs for 113 long intergenic lincRNAs using the *nmt1* promoter (*Maundrell, 1993*). The full-length lincRNA sequences, as annotated in PomBase (*Lock et al., 2019*), were amplified by PCR using the high-fidelity Phusion DNA polymerase (NEB) and cloned into the pJR1-41XL vector (*Moreno et al., 2000*) using the CloneEZ PCR Cloning Kit (GenScript). All primer sequences used for cloning are provided in *Supplementary file 1*. Each plasmid was checked by PCR for correct insert size. We also checked inserts of the overexpression plasmids by Sanger sequencing in a 96-well plate format (Eurofins Genomics) using a universal forward primer (5′ CGGA TAATGGACCTGTTAATCG 3′) for the pJR1-41XL plasmid upstream of the cloning site. This HTP sequencing produced reliable sequence data for 80 inserts, including full insert sequences for (62 plasmids and the first ~900 bp of inserts for 18 plasmids). Of these, only the insert for *SPNCRNA.601* showed a sequence error compared to the reference genome, a T to C transition in position 559. Plasmids were transformed into *S. pombe* cells (h⁻, *leu1-32*), and leucine prototroph transformants were selected on solid Edinburgh Minimal Medium (EMM2) plates. An empty-vector control strain was created analogously. Of the 113 lincRNAs, 67 were represented by two independently cloned vectors (*Supplementary file 1*).

For eight lincRNA overexpression constructs and the empty-vector control strain, we carried out RT-qPCR assays to determine the expression of selected overexpressed lincRNAs relative to their native genomic expression (*Figure 6—figure supplement 1A*). For this, cells were grown in EMM2 to an $OD_{600}$ of ~0.5. RNA was extracted with TRIzol reagent (Invitrogen) according to the manufacturer's protocol, followed by DNase digestion (Invitrogen, Turbo DNase). RNA (1 µg) was reverse transcribed with SuperScript III reverse transcriptase and random hexamers (Invitrogen), according to the manufacturer's recommendations. Then, 4 µl of a 1:5 dilution of the resulting cDNA was used to quantify the transcripts for a 10 µl reaction with 5 µl of Fast SYBR Green Master Mix (Applied Biosystems) and 250 nM of each primer in a QuantStudio 6 Flex instrument (Applied Biosystems) in the fast cycling mode, according to the manufacturer's recommendations. Transcript levels of all samples were normalized to *act1*, and final lincRNA transcript levels were calculated relative to the empty-vector control. All primers used for the assay are listed in *Supplementary file 1*.

## HTP phenotyping of deletion mutants on solid media

The deletion mutants were broadly phenotyped using a colony-based phenomics platform as described (*Kamrad et al., 2021*; *Kamrad et al., 2020b*). Mutants were assayed on solid media with a variety of 55 unique stressors using different concentrations and, in some cases, combinations of stressors. In total, we assayed 134 different conditions, with the viability dye phloxine B being included in 66 of these conditions. *Figure 1—figure supplement 1* provides a description of the conditions used for phenotyping. *Supplementary file 1* contains the concentrations of all the stressors used. Cells were grown for 24 hr on yeast extract supplement (YES) plates in 384-colony format containing a wild-type control grid, followed by pinning cells onto plates containing the stressors using reduced pressure (4% pinning pressure to transfer a small amount of biomass). Plates were wrapped in plastic to avoid drying out and incubated for ~40 hr at 32°C, unless stated otherwise, before image acquisition and phenotype assessment. Deletion strains were assayed with at least three independent biological repeats using two or more colonies (technical repeats) for each biological repeat. In most cases, we had two or more independently generated deletion strains for each lincRNA (using the same or different gRNAs), and we performed at least three biological repeats for each strain. The numbers of independent strains for each lincRNA are provided in *Supplementary file 1* (sheet: lincRNA_metadata; column: n_independent_ko_mutants). The total numbers of repeats carried out for each condition after QC filtering are available in *Supplementary file 2* (columns: observation_count).

Image acquisition and quantitation, data normalization and processing, as well as hit calling were performed using our *pyphe* pipeline, which is available here: https://github.com/Bahler-Lab/pyphe (*Kamrad et al., 2020b*). Images of plates were acquired with a flatbed scanner (Epson V800 Photo), controlled by *pyphe-scan* through SANE. Images for quantifying colony area (growth) were taken by transmission scanning using the --*mode Grey* argument. For quantifying redness/viability, images were taken by reflective scanning using --*mode Color*. Images were acquired at 300 or 600 dpi resolution. For colour images, to determine colony redness for viability, we used an opaque fixture to hold

the plates in place, the white cover was installed in the scanner lid, and the scanner was covered by a cardboard box to prevent external light interfering with image acquisition. Images were inspected individually and excluded if one or more of the following applied: several colonies were missing due to pinning errors (usually in the corners), evidence of contamination, white background had not been inserted during colour scanning, and/or plate had slipped significantly during scanning so that a whole row/column of colonies was missing from the image. The overall number of excluded plates was low and generally did not result in significant data loss in the final dataset due to the large number of replicate plates. For image quantification, greyscale transmission images for colony area quantitation were analysed with the R package *gitter* (*Wagih and Parts, 2014*) using the following parameters: *plate.format = 384, inverse="TRUE", remove.noise="TRUE", autorotate="TRUE"*. Images for which *gitter* failed (very few) were excluded from further analysis. Colour images for redness/viability quantification were analysed with *pyphe-quantify* using default parameters.

For data normalization and processing, an experimental design table was prepared for each dataset which listed for each plate the path to the data file produced during image quantification, plate layout information, the condition, as well as other metadata (e.g. batch number, replicate counter, and free-text comments). Data from all images of the same dataset were parsed and processed simultaneously using *pyphe-analyse*, producing a single data report table in tidy format per experiment, containing all data associated with a single measured colony on each line. For analysis of colony areas, the following parameters were used: *--format gitter --load_layouts --gridnorm standard384 --rcmedian --check*. For colony redness analysis, the options were *-format pyphe-quantify-redness --load_layouts --rcmedian --check*.

*pyphe* performs some automated quality control. Specifically, during grid normalization, missing reference grid colonies are flagged and all neighbouring colonies are set to NA. *pyphe* also checks data for negative and infinite fitness values (rare artefacts of normalization procedures). For the colony size datasets, additional quality control of the data was performed as follows: missing colonies (colony size 0 reported by gitter and fitness 0 reported by *pyphe-analyse*) were set to NA as these are pinning errors; colonies with a circularity (reported by *gitter*) below 0.85 were set to NA; plates with a coefficient of variation (CV) of >0.2 for wild-type controls were set to NA. For viability datasets, the only QC step was to exclude plates with a wild-type CV of >0.05.

For statistical analysis, tables reporting summary statistics and p-values for each lincRNA gene and condition were obtained with *pyphe-interpret*. Hits were called separately for control conditions (where we tested for difference in means between each lincRNA mutant and wild-type control in the same condition) and all other conditions (where we tested for difference in means between each lincRNA mutant in test condition vs. corresponding control condition). Welch's *t*-test, which does not assume homogeneity of variances, was used, and the obtained p-values were corrected for multiple testing for each condition separately using the Benjamini–Hochberg method (*Benjamini and Hochberg, 1995*).

The dataset for clustering (*Supplementary file 5*, *Figure 5*) was derived from *Supplementary files 2 and 4* by subtracting one from the MES and dividing by the standard deviation of the wild-type control for each condition. Conditions were then aggregated by choosing the strongest response across all repeats of the same stressor (the stressor is indicated in the 'stress_description' column in the knock-out_condition_metadata sheet of *Supplementary file 1*). As not all lincRNA mutants were phenotyped in all conditions, clustering was restricted to a set of 41 core stressors. lincRNA or coding-gene mutants with less than five responses were excluded, leaving 194 mutants in total, including 16 ncRNAs. The final dataset only contained 17 NA values which were imputed with 0. Hierarchical clustering was done with *scipy* (*Virtanen et al., 2020*) using the Ward method and the Euclidean distance metric. Clusters were obtained by cutting the dendrogram using the fcluster function with the 'maxclust' method. Functional enrichments in clusters 1–3 were analysed using AnGeLi (*Bitton et al., 2015*), with all protein-coding genes as background list.

Data for the phenotypic correlation network (*Figure 5B*) were generated from phenotypes for all lincRNA and coding-gene deletions using a ternary system: resistant, sensitive, and no phenotype encoded as 1, –1, and 0, respectively. The network was generated following general instructions (*Contreras-López et al., 2018*; *Shannon et al., 2003*). Briefly, we used Pearson correlations to calculate the network and filtered on absolute *r* values above 0.6 and adjusted p<0.01. Clustering of the network in Cytoscape was done using community clustering (GLay) from the clustermaker extension (*Morris et al., 2011*).

## HTP microscopy and flow cytometry for cell-size and cell-cycle phenotypes

Strains, frozen in glycerol in 384-colony format, were revived in YES solid plates, resuspended into 150 µl of liquid YES in 96-well plates and incubated at 32°C for 16 hr. Then, 100 µl from these pre-cultures were added to 1.5 ml of preheated (32°C) liquid YES in 96 deep-well plates and incubated at 32°C for 8 hr. Cells were collected by centrifugation, cell pellets were resuspended in 70% ice-cold ethanol, and stored in the dark at 4°C until further processing. As cell-size and cell-cycle phenotype controls, we used two temperature-sensitive cell-cycle mutants: *cdc10-129* and *wee1-50*. These mutants were grown in 50 ml YES at 25°C, centrifuged, and resuspended in 50 ml of prewarmed (37°C) YES and incubated for 4 hr at 37°C to block cell-cycle progression. After 4 hr, 1 ml of the samples was fixed for microscopy and flow cytometry. The remaining cells were centrifuged and resuspended in 50 ml of prewarmed YES (25°C), incubated at 25°C, and samples collected and fixed after 20 and 60 min. Over 80% of the 110 lincRNA mutants screened for cellular phenotypes were assayed in at least two independent biological repeats.

For cell-size and cell-cycle phenotypes, fixed cells were washed in 50 mM sodium citrate buffer, spun down at 3000 × *g* for 5 min, resuspended in 50 mM sodium citrate containing 0.1 mg/ml RNAse A, and incubated at 37°C for 2 hr. Cells were then spun down at 3000 × *g* for 5 min and resuspended in 500 µl of 50 mM sodium citrate +1 µM SYTOX Green (Thermo Fisher Scientific, cat. # S7020). Immediately prior to analysing samples using either HTP flow cytometry or HTP microscopy, cells in the deep well plates were sonicated for 40 s at 50 W (JSP Ultrasonic Cleaner model US21) to increase the efficiency of singlets.

For HTP image acquisition, cells were further stained with a 1:1000 dilution of CellMask Deep Red Plasma membrane dye (Thermo Fisher Scientific, cat. # C10046), according to the manufacturer's instructions. Then, 2.5 µl of fixed and stained cells were transferred from 96-well plates into a poly-lysine-coated 384-well Perkin Elmer Cell Carrier Ultra imaging plate (PerkinElmer, cat. # 6057500), pre-filled with 25 µl of 1 µM SYTOX Green using a Biomek Fx robot. Cells were spun down for 3 min at 200 × *g* before imaging. Imaging was performed on a Perkin Elmer Opera Phenix microscope using a water immersion 63× lens to capture confocal stacks of 12 planes in both Alexa488 (SYTOX Green) and Alexa647 (CellMask) channels, with 63 microscopic fields being captured per sample. The images were projected and analysed using the associated Phenix software Harmony for the automated identification of mono- and binucleated cells and respective cell length. Features were exported for further analysis using R studio.

For HTP flow cytometry, 250 µl of cells were transferred into 96-well plates and 30,000 cells were measured in a Fortessa X20 Flow cytometer (BD Biosciences) using the HTS plate mode on the DIVA software and a 488 nm excitation laser to capture the SytoxGreen DNA staining. Populations of interest were gated as described (*Knutsen et al., 2011*) using the FlowJo software version 10.3.0. Features of interest (populations with different DNA content) were then exported for further analysis using R studio. The determined percentage of cells in each cell-cycle phase per sample was used to validate the HTP imaging data. For correlation with the HTP imaging (binucleated cells), S- and G1-phase cell populations were grouped together (*Supplementary file 2*).

Data analysis was carried out in R (v.3.5.2), using the package tidyverse for data manipulation, visualization, and statistical analysis. All tests were two-sided unless otherwise stated. For HTP imaging analysis, cell density was checked for each sample in the multi-well plate and given a score of 0–5, where 0 is very low to no density (<50 cells/well) and 5 is at too high density; samples scoring 0 and 5 were excluded from analysis. For cell-size analysis, the median cell size of binucleated cells for each mutant was used to calculate fold-changes relative to wild-type values, applying the Wilcoxon test to determine significant differences (p<0.05), only considering cells showing a ≥ 5% difference in size compared to wild-type cells. For cell-cycle analysis by HTP imaging, the percentage of binucleated cells per microscopic field (63 fields/sample) was used to calculate the median value per lincRNA mutant, followed by fold-change analysis calculated by normalizing the percentage of binucleated cells in each sample relative to wild-type values, applying the Wilcoxon test to determine significant differences (p<0.05), only considering cells showing a ≥20% difference in binucleated cells compared to wild-type.

## HTP phenotyping of overexpression mutants on solid media

Overexpression strains were arrayed in 384 format together with the empty-vector control strain and a grid of the wild-type strain (*972 h-*) for normalization. Strains were revived from glycerol stocks in YES and grown for 2 days at 32°C. Colonies were then transferred to new YES plates, grown for 1 day, and pinned onto EMM2 (with $NH_4Cl$ but without amino acid supplements) with or without the specified drugs/supplements. YES medium contains thiamine that represses the *nmt1* promoter and leucine that compromises the maintenance of the overexpression plasmid (which contains the *LEU2* marker). We screened the lincRNA overexpression library for colony growth phenotypes in 47 conditions (*Supplementary files 1 and 6*). Each overexpression strain was represented by at least 12 colonies across three different plates and experiments were repeated at least three times. Each condition was assayed in three independent biological repeats, together with control EMM2 plates, resulting in at least 36 data points per strain per condition. Plates were incubated at 32°C if not stated otherwise for the condition. Plates were imaged as described for deletion mutants after 40 or 64 hr in order to capture as many hits as possible.

Image acquisition and quantification, data normalization and processing, as well as hit calling were performed using the *pyphe* pipeline as described above for greyscale transmission images to quantify colony sizes. During grid correction, 24,683 colonies were excluded due to missing grid colonies and 2539 missing colonies were set to 'NA' (pinning errors), and data from 290 of 2772 plates were discarded because they either showed a fraction of unexplained variance (FUV) above 1 or a control CV of >0.5. The final dataset contained 917,368 data points. The colonies which passed the above quality control steps were normalized with the grid first, and the resulting colony sizes were additionally normalized to the control condition (EMM2) for the conditions with stressors. All data from the *pyphe* analysis are provided in *Supplementary file 6*. The hits were defined by adjusted (Benjamini–Hochberg) p-values ≤ 0.01 and MES ≥5% compared to empty-vector control.

For clustering analyses, we first filtered the relative log2 MES data (relative to empty-vector control) for genes with five or more hits followed by conditions with five or more hits, resulting in 59 lincRNA mutants with MES data for 29 conditions. Then we discretized the data, classing mutants as either sensitive (–1), resistant (+1), or similar to their fitness in the corresponding control condition (0). We performed hierarchical clustering with the complete method using the Canberra distance metric and plotted the heatmap (*Figure 6B*) with the ComplexHeatmap r-package (*Gu et al., 2016*).

For correlation analyses between deletion vs. overexpression data, we filtered the phenotyping data for the 104 shared lincRNA mutants and the 22 shared stress conditions between the two mutant types. As the overexpression strains could only be assayed on minimal media while the deletion strains were mainly assayed on rich media, we matched conditions based on the added drug/stressor only, disregarding the media background. In case of multiple related conditions (e.g. same stress in different doses), the strongest response was used (maximum MES).

## Phenotyping of meiotic differentiation for selected lincRNA mutants

We used CRISPR/Cas9 deletion mutants in the homothallic $h^{90}$ background for the lincRNAs in the sub-cluster of *Figure 6B*: *SPNCRNA.335, SPNCRNA.1154, SPNCRNA.1530*, and *meiRNA*, along with a wild-type $h^{90}$ control strain (968). The strains were grown in liquid YES medium at 32°C to an $OD_{600}$ of ~0.5. Cells were washed 3× in EMM-N medium and resuspended in EMM-N to an $OD_{600}$ of ~1.5. Cultures were then grown at 25°C in a shaking incubator at 180 rpm.

To assess meiotic differentiation, 1 ml samples were harvested at 24 and 72 hr after the medium change, centrifuged, and pellets resuspended in 70% ethanol at 4°C. For microscopy, 100 µl of ethanol-fixed cells were rehydrated in 50 mM sodium citrate and incubated overnight at 4°C. Cells were centrifuged for 5 min at 1500 × *g*, the pellets were resuspended in 10 µl of 50 mM sodium citrate, and 2 µl of cell suspension was spread onto a slide and mounted with DAPI containing mounting media. Cells were imaged using a Zeiss ApoTome.2 microscope with a Hamamatsu digital camera and 63× objective. At least 500 cells per sample were counted using Fiji (*Schindelin et al., 2012*) by differentiating between cells, zygotes, asci, or free spores. Mating efficiency was determined as described (*Rodríguez-Sánchez et al., 2011*), calculated as {[2× (number of zygotes + number of asci)] + ½ number of spores} divided by {[2× (number of zygotes + number of asci)]+ ½ number of spores + number of non- mating cells}.

To measure spore viability (*Escorcia and Forsburg, 2018*), 5 ml samples were harvested at 72 hr after the medium change, centrifuged, resuspended in 1 ml 0.5% glusulase solution (1:10 dilution of 5% glusulase in sterile $H_2O$), and incubated for ~16 hr at 25°C. These glusulase suspensions were diluted at 1:5 to 1:20 in sterile $H_2O$, and spore numbers were counted using a hemocytometer. For each strain, ~200 spores were plated onto three YES-agar plates and incubated at 32°C until colonies formed. The colony numbers were counted and divided by the number of spores plated to determine the proportion of viable spores. This entire procedure was repeated in three independent biological replicates.

## Acknowledgements

We thank Sarah Aldous, Natalia Borbarán Bravo, Mike Jiang, Janina Karl, Patrycja Krawczyk, Sara Kurz, Kira Kyne, Shirley Liu, and Florence Young for help with deletion and overexpression constructs. This work was supported by a Wellcome Trust Senior Investigator Award (095598/Z/11/Z) and BBSRC Research Grants (BB/R018219/1 and BB/R009597/1) to JB. ST was supported by a Boehringer Ingelheim Fonds PhD Fellowship. The Francis Crick Institute receives its core funding from Cancer Research UK (FC001134), the UK Medical Research Council (FC001134), and the Wellcome Trust (FC001134).

## Additional information

### Funding

| Funder | Grant reference number | Author |
| --- | --- | --- |
| Wellcome | 095598/Z/11/Z | Jürg Bähler |
| Biotechnology and Biological Sciences Research Council | BB/R018219/1 | Jürg Bähler |
| Biotechnology and Biological Sciences Research Council | BB/R009597/1 | Jürg Bähler |
| Boehringer Ingelheim Fonds | | StJohn Townsend |
| Wellcome Trust Centre for Mitochondrial Research | FC001134 | Markus Ralser |

The funders had no role in study design, data collection and interpretation, or the decision to submit the work for publication.

### Author contributions

Maria Rodriguez-Lopez, Conceptualization, Formal analysis, Investigation, Methodology, Supervision, Validation, Visualization, Writing - original draft, Writing – review and editing; Shajahan Anver, Cristina Cotobal, Formal analysis, Investigation, Methodology, Validation, Visualization, Writing – review and editing; Stephan Kamrad, Data curation, Formal analysis, Investigation, Software, Visualization, Writing – review and editing; Michal Malecki, Conceptualization, Formal analysis, Investigation, Methodology, Supervision, Writing – review and editing; Clara Correia-Melo, Investigation, Methodology, Visualization, Writing – review and editing; Mimoza Hoti, Investigation, Project administration; StJohn Townsend, Formal analysis, Software, Writing – review and editing; Samuel Marguerat, Funding acquisition, Methodology, Supervision, Writing – review and editing; Sheng Kai Pong, Investigation; Mary Y Wu, Methodology; Luis Montemayor, Validation; Michael Howell, Methodology, Supervision; Markus Ralser, Funding acquisition, Supervision, Writing – review and editing; Jürg Bähler, Conceptualization, Formal analysis, Funding acquisition, Project administration, Supervision, Writing - original draft

### Author ORCIDs

Maria Rodriguez-Lopez http://orcid.org/0000-0002-2066-0589
Shajahan Anver http://orcid.org/0000-0002-7582-5125
Stephan Kamrad http://orcid.org/0000-0002-5957-4661

Michal Malecki [iD] http://orcid.org/0000-0002-1525-5036
Clara Correia-Melo [iD] http://orcid.org/0000-0001-6062-1472
Samuel Marguerat [iD] http://orcid.org/0000-0002-2402-3165
Sheng Kai Pong [iD] http://orcid.org/0000-0002-9940-1175
Mary Y Wu [iD] http://orcid.org/0000-0002-2074-6171
Michael Howell [iD] http://orcid.org/0000-0003-0912-0079
Jürg Bähler [iD] http://orcid.org/0000-0003-4036-1532

**Decision letter and Author response**
Decision letter https://doi.org/10.7554/eLife.76000.sa1
Author response https://doi.org/10.7554/eLife.76000.sa2

## Additional files

### Supplementary files

• Supplementary file 1. Details of strains and conditions.

• Supplementary file 2. Phenotype data of long intergenic non-coding RNA (lincRNA) deletion mutants.

• Supplementary file 3. Summary tables of long intergenic non-coding RNA (lincRNA) and coding-gene deletion mutants.

• Supplementary file 4. Phenotype data of coding-gene deletion mutants.

• Supplementary file 5. Clustering analysis of deletion-mutant phenotypes.

• Supplementary file 6. Phenotype data of long intergenic non-coding RNA (lincRNA) overexpression mutants.

• Transparent reporting form

### Data availability

Data generated or analysed during this study are included in the manuscript and supporting files.

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
