## [Decision Letter]

[Editors' note: this paper was reviewed by Review Commons.]

---

## [Author Response]

Reviewer #1 (Evidence, reproducibility and clarity (Required)):The manuscript by Rodriguez-Lopez et al. describes the analysis of long intergenic noncoding RNA (lincRNA) function in fission yeast using both deletion and overexpression methods. The manuscript is very well presented and provides a wealth of lincRNA functional information for the field. This work is an important advance as there is still very little known about the function of lincRNAs in both normal and other conditions. An impressive array of conditions were assessed here. With a large scale analysis like this there is really not one specific conclusion. The authors conclude that lincRNAs exert their function in specific environmental or physiological conditions. This conclusion is not a novel conclusion, it has been proposed and shown before, but this manuscript provides the experimental proof of this concept on a large scale.The lincRNA knock-out library was assessed using a colony size screen, a colony viability screen and cell size and cell cycle analysis. Additionally, a lincRNA over-expression library was assessed by a colony size screen. These different functional analysis methods for lincRNAs were than carried out in a wide variety of conditions to provide a very large dataset for analysis. Overall, the presentation and analysis of the data was easy to follow and informative. Some points below could be addressed to improve the manuscript.There were 238 protein coding gene mutants assessed in parallel, to provide functional context, which was a very promising idea. But, unfortunately, the inclusion of 104 protein coding genes of unknown function restricted the use of the protein coding genes in the integrated analysis to connect lincRNAs to a known function using guilt by association.

Yes, the unknown coding-gene mutants did certainly not help to provide functional context through guilt by association. These mutants were included to generate functional clues for the unknown proteins and compare phenotype hits with unknown lincRNA mutants. Nevertheless, because the known coding-gene mutants included broadly cover all high-level biological processes (GO slim), we could make several useful functional inferences for certain lincRNAs as discussed.

The colony viability screen is not described well throughout the manuscript. Firstly, the use of phloxine B dye to determine cell viability needs to be described better when first introduced at the bottom of page 6. What exactly is this viability screen and red colour intensity indicating? Please define what the different levels of red a colony would indicate as far as viability. I assume an increase in red colour indicates more dead cells? So it is confusing that later the output of this assay is described as giving a resistant/sensitive phenotype or higher/lower viability. How can you get a higher viability from an assay that should only detect lower viability? Shouldn't this assay range from viable (no, or low red, colour) to increasing amounts of red indicating increasingly less viability? Figure 4D is also confusing with the "red" and "white" annotations. These should be changed to "lower viability" and "viable" or "not viable" and "viable".

The colony-viability screen is described in detail in our recent paper (Kamrad et al., *eLife* 2020). We have now better explained how phloxine B works to determine cell viability (p. 6). The reviewer’s assumption is correct: an increase in red colour indicates more dead cells. However, all phenotypes reported are relative to wild-type cells under the same condition. Many conditions lead to a general increase in cell death, but some mutants show a lower increase in cell death compared to wild-type cells. These mutants, therefore, have a higher viability than wild-type cells, i.e. they are more resistant than wild-type under the given condition. We have tried to clarify this in the text, including the legend of Figure 4. We agree that the ‘red’ and ‘white’ annotations in Figure 4D could be confusing. We have now changed these to ‘low viability’ and ‘high viability’. Again, this is relative to wild-type cells.

How are you sure that when generating the 113 lincRNA ectopic over-expression constructs by PCR that the sequences you cloned are correct? Simply checking for "correct insert size", as stated in the methods, is not really good practice and these constructs should be fully sequenced to be sure they contain the correct sequence and that constructs have not had mutations introduced by the PCR used for cloning. Without such sequence confirmation one cannot be completely confident that the data produced is specific for a lincRNA overexpression. Additionally, a selection of strains with the overexpression constructs should be tested by qRT-PCR and compared to a non-over-expressing strain to confirm lincRNA overexpression.

To minimize errors during PCR amplification, we used the high-fidelity Phusion DNA polymerase which features an >50-fold lower error rate than Taq DNA Polymerase. We had confirmed the insert sequences for the first 17 lincRNAs cloned using Sanger sequencing (but did not report this in the manuscript). We have now checked additional inserts of the overexpression plasmids by Sanger sequencing in 96-well plate-format using a universal forward primer upstream of the cloning site. This high-troughput sequencing produced reliable sequence data for 80 inserts, including full insert sequences for 62 plasmids and the first (~900 bp of insert sequences for 18 plasmids). Of these, only the insert for *SPNCRNA.601* showed a sequence error compared to the reference genome: T to C transition in position 559. This mutation could reflect either an error that occurred during cloning or a natural sequence variant among yeast strains (lincRNA sequences are much more variable than coding sequences). So, in general, the PCR cloning accurately preserved the sequence information. We have added this information in the Methods (p. 27-28). Please note that lincRNAs depend much less on primary nucleotide sequence than mRNAs, and a few nucleotide changes are highly unlikely to interfere with lincRNA function.

Minor comments:Page 4, lines 19-20 – "A substantial portion of lincRNAs are actively translated (Duncan and Mata, 2014), raising the possibility that some of them act as small proteins." This sentence does not make sense, lincRNAs can't "act as" small proteins, they can only "code for" small proteins. Wording needs to be changed here.

We agree and have changed the wording as suggested.

Figure 1A is a nice representation but what are the grey dots? Are they all ncRNAs including lincRNAs? This needs to be stated in the legend.

The grey dots represent all non-coding RNAs across the three *S. pombe* chromosomes as described by Atkinson et al., 2018. This has now been clarified in the legend.

How many lincRNAs are there in total in pombe and what percentage did you delete? These numbers should be stated in the text.

There are 1189 lincRNAs and we mutated ~12.6% of them. These numbers are now stated at the end of the Introduction, page 5.

It would be nice if Supplementary Figure 1 included concentrations or amounts of the conditions used. This info is buried in a Supplementary table and would be better placed here.

Supplemental Figure 1 provides a simple overview for the different conditions and drugs used. For most stresses and drugs, we used multiple different doses. So the figure would become cluttered if we indicated all these concentrations, detracting from the main message. Colleagues who are interested in the different concentration ranges used for specific conditions can readily obtain this information from Supplemental Dataset 1. We have now added a statement in this respect to the legend of Supplemental Figure 1*.*

Page 6, last sentence. What is a "biological repeat"? Three distinct deletion strains (ie three different deletion strains made by CRISPR) or one deletion strain used three times?

Biological repeat means that one deletion strain was assayed three times independently, each with at least two colonies (technical repeats). In most cases, we had two or more independently generated deletion strains for each lincRNA (using the same or different gRNAs), and we performed at least three biological repeats for each strain. The numbers of independent strains for each lincRNA are provided in Supplemental Dataset 1 (sheet: lincRNA_metadata, column: n_independent_ko_mutants). The total numbers of repeats carried out for each condition after QC filtering are available in Supplemental Dataset 2 (columns: observation_count). We have clarified this on p. 7, and the details are now provided in the Methods on p. 28-29 (deletion mutants) and p. 32 (overexpression mutants).

There is no mention in the manuscript of how other researchers can get access to the deletion strains and over-expression plasmids.

As is usual, all strains and plasmids will be readily available upon request.

Reviewer #1 (Significance (Required)):The production of lincRNA deletion strains and overexpression plasmids, and their analysis under an impressive number of conditions, provides key resources and data for the ncRNA field. This work complements nicely the analysis of protein coding gene deletion strains and provides the tools and data for future mechanistic studies of individual lincRNAs. This work would be of interest to the growing audience of ncRNA researchers in both yeast and other systems.Field of expertise:Yeast deletion strain construction and analysis, RNA functional analysisReferee Cross-commenting:Reviewer #3 makes an important point that the stability of each lincRNA over expressed from plasmid is not known and therefore some lincRNAs may not be overexpressed as predicted. RT-qPCR would be required to assess lincRNA expression levels from the plasmids. It also appears that we both agree that it is important to determine the sequence of the cloned lincRNAs in the over expression plasmids.

See reply in response to Reviewer 3.

Reviewer #3 also makes an important point in his review that where it is predicted that a lincRNA deletion influences an adjacent gene in cis then the expression of that gene should be tested.

See reply in response to Reviewer 3.

Reviewer #2 (Evidence, reproducibility and clarity (Required)):Summary:The Rodriguez-Lopez manuscript from the Bahler lab present the phenotypical and functional profiling of lincRNA in fission yeast. This is the first large-scale, extensive work of this nature in this model organism and it therefore nicely complement the well-documented examples of lincRNA already reported in *S. pombe*.The work is very solid using seamless genome deletion and overexpression followed by colony-based assay in respone to a very wide set of conditions.Major comments:Considering that this is a descriptive work by nature and that the experiments were properly conducted as far as I can judge, I don't have major issues with this paper.To me the only thing that is missing is a gametogenesis assay, for two reasons: First, several reported cases of lincRNAs in pombe critically regulates meiosis, and second many of the analysed lincRNAs are upregulated durig meiosis. Figure 6B already points to three obvious candidates. I don't think it would take to much time to look at the deletion and OE in an h90 strain and see the effect of gametogenesis for the entire set or at least the 3 candidates from Figure 6.If the already broad set of lincRNAs implicated in meiosis would grow, this would be another evidence that eukaryotic cell differentiation relies on non-coding RNAs even in simpler models.

We agree that this is a meaningful analysis to add. We have now deleted the three unstudied lincRNA genes, along with the meiRNA gene, from the sub-cluster of Figure 6B in the homothallic *h^90^* background (to allow self-mating). We have analysed meiosis and spore viability of these four deletion strains together with a wild-type *h^90^* control strain. These experiments indicate that cell mating is normal in the deletion mutants, but meiotic progression is somewhat delayed in *SPNCRNA.1154*, *SPNCRNA.1530* and, most strongly, *meiRNA* mutants (the latter has been reported before (reviewed by Yamashita 2019)). Notably, we detected significant reductions in spore viability for all four deletion mutants compared to the control strain. These results point to roles of *SPNCRNA.1154*, *SPNCRNA.1530*, and *SPNCRNA.335* in meiotic differentiation, as predicted by the clustering analyses. This is a nice addition to the manuscript. We now report these results on p. 23, with a new Supplemental Figure 10, and describe the experimental procedures in the Methods (p. 34-35).

Minor comments:
*– A reference to the recent work of the Rougemaille lab on mamRNA is necessary*

Reply: Yes, we now cite this reference in the Introduction (p. 4).

– A discussion of the possibility to perform large-scale genetic interactions searches (as done by Krogan for protein-coding genes) would add to the discussion of future plans.

We have added a sentence about the potential of SGA screens in the Conclusions

(p. 26).

Reviewer #2 (Significance (Required)):The work unambiguously shows that most of the lincRNAs analyzed exert cellular functions in specific environmental or physiological contexts. This conclusion is critical because the biological relevance this so-called « dark matter » is still debated despite a few well-established cases. This is an important addition to the field and the deep phenotyping work already points to some directions to analyse some of these lincRNA in the context of cell cycle progression, metabolism or meiosis.Referee Cross-commenting:– I agree with the issues raised by referees 1 and 3 but I am concerned about the added value of a RT-qPCR. First, this is a significant amount of work considering the large set of targets. Second a more importantly, what you’ll end up with is a fold change. What will be considered as overexpression? Which threshold? This is why I prefer a biological read-out (a phenotype) because whatever the fold change, it tells us that there is an effect. It is very likely indeed that some targets are not overexpressed because of their rapid degradation. To me, this is the drawback of any large-scale studies.– Also, looking at the expression of the adjacent gene in the case of a cis-effect is interesting though this is likely condition-dependent (because most phenotypes appear in specific conditions). So, what would be the conclusion if there is no effect in classical rich media?– The sequence of the insert should be specified, I agree. Most likely, it is the sequence available from pombase (this is what I understood) but that should be clarified indeed.

Yes, the sequences of the inserts are available from PomBase, and we provide the primer sequences used for cloning in the Supplemental Dataset 1. We have now clarified this in the Methods (p. 27).

Reviewer #3 (Evidence, reproducibility and clarity (Required)):In this work from the group of Jurg Bahler, the authors take advantage of the high throughput colony-based screen approach they recently developed (Kamrad et al., eLife 2020) to perform a functional profiling analysis on a subset of 150 lincRNAs in fission yeast. Using a seamless CRISPR/Cas9-based method, they created deletion mutants for 141 lincRNAs. In addition, the authors also generated strains ectopically overexpressing 113 lincRNAs from a plasmid (under the control of the strong and inducible nmt1 promoter).The viability and growth of all these mutants was then assessed across benign, nutrient, drug and stress conditions (149 conditions for the deletion mutants, 47 conditions for the overexpression). For the deletion mutants, the authors also assayed in parallel mutants of 238 protein-coding genes (PCGs) covering multiple biological processes and main GO classes.In benign conditions, deletion of 5 and 10 lincRNAs resulted in a reduced growth phenotype (rich and minimal medium, respectively). Morphological characterization by microscopy also revealed cell size defects for 6 lincRNA mutants (2 shorter, 4 longer). In addition, 27 mutants displayed phenotypes pointing defects in the cell cycle.Remarkably, the nutrient/drug/stress conditions revealed more phenotypes, with 60 of the 141 lincRNA mutants showing a growth phenotype in at least one condition, and 25 mutants showing a different viability compared to the wild-type in at least one condition.Also remarkable is the observation that 102/113 lincRNA overexpression strain displayed a growth phenotype in at least one condition, 14 lincRNAs showing phenotypes in more than 10 conditions.The clustering analyses performed by the authors also provide functional insight for some lincRNAs.Overall, this is an important study, well conducted and well presented. Together, the data described by the authors are convincing and highlight that most lincRNAs would function in very particular conditions, and that deletion/inactivation and overexpression are complementary approaches for the functional characterization of lncRNAs. This has been demonstrated here, in a very elegant manner.I think this manuscript will be acknowledged as a pioneer work in the field.Major comments:1. Are the key conclusions convincing?To my opinion, the key conclusions of this study are convincing.2. Should the authors qualify some of their claims as preliminary or speculative, or remove them altogether?No. The authors are careful in their claims and conclusions.3. Would additional experiments be essential to support the claims of the paper? Request additional experiments only where necessary for the paper as it is, and do not ask authors to open new lines of experimentation.This study is based on systematic lincRNA deletion/overexpression.– For the deletion strains, I could not find any information about the control of the deletions. Are the authors sure that the targeted lincRNAs were indeed properly deleted?

Yes, we had carefully checked the correctness of the deletions using several controls as described by Rodriguez-Lopez et al. 2017. All deletion strains were checked for missing open-reading frames by PCR. For 20 strains, we also sequenced across the deletion scars. We re-checked all strains by PCR after arraying them onto the 384 plates to ensure that no errors occurred during the process. We have now specified this in the Methods (p. 27).

– For the overexpression, there is only a control of the insert size by PCR. Sanger sequencing would have been preferable to confirm that the targeted lincRNAs were properly cloned, without any mutation. In addition, the authors did not check that the lincRNAs were indeed overexpressed (at least in the benign conditions). Is the overexpression fold similar for all the lincRNAs? Do the 14 lincRNAs showing the most consistent phenotypes in at least 10 conditions display different expression levels than the other lincRNAs?4. Are the suggested experiments realistic in terms of time and resources? It would help if you could add an estimated cost and time investment for substantial experiments.– Validating the deletion strains requires genomic DNA extraction and then PCR. This is repetitive and tedious, but this control is important, I think. The time needed depends on the possibility of automating the process. I think this is feasible in this lab.– Controlling the insert sequence into the overexpression vector requires plasmid DNA (available as it was used for PCR) and one/several primer(s), depending on the insert size. The sequencing itself is usually done by platforms.– Analysing lincRNA overexpression at the RNA level requires yeast cultures, RNA extraction and then RT-qPCR. Again, the time needed depends on the possibility of automating the process.

We have now checked most overexpression constructs by Sanger sequencing of the inserts as described in response to Reviewer 1. Moreover, we have tested the overexpression levels for eight selected overexpression constructs using RT-qPCR analysis. These eight constructs feature the entire range of associated phenotypes hits, including 3 lincRNAs with the highest number of phenotypes in 14 conditions, 3 with no phenotypes, and 2 with intermediate numbers of phenotypes. The RT-qPCR results show that the lincRNAs were 35- to 2200-fold overexpressed relative to the empty-vector control strain (which expresses the lincRNA at native levels). No clear pattern was evident between expression levels and phenotype hits, e.g. lincRNAs without phenotypes when overexpressed showed similar fold-changes as a lincRNA showing 13 phenotypes. We present these results on p. 21/22 and in the new Supplemental Figure 9A, and describe the experiment in the Methods (p. 28).

As pointed out by Reviewer 2, these fold changes in expression are actually of limited value compared to the phenotype read-outs. The important result is that we detected phenotypes for over 90% of the overexpression strains, indicating that overexpression generally worked. Given that this is a large-scale study, there might be some lincRNA constructs that are faulty or are not overexpressed. It would not be realistic or meaningful to test all constructs. Any follow-on studies focusing on a specific lincRNAs will need to first validate the large-scale results as is common practice.

5. Are the data and the methods presented in such a way that they can be reproduced?The methods are clearly and extensively explained. If necessary, the reader can find more details about the high-throughput colony-based screen approach in the original paper (Kamrad et al., eLife 2020); a very interesting technical discussions can also be found in the reviewers reports and in the authors response published alongside.6. Are the experiments adequately replicated and statistical analysis adequate?The experiments are replicated. However, I feel confused regarding the number of replicates used in each analysis.In the first part of the Results, it is mentioned that all colony-based phenotyping was performed in at least 3 independent replicates, with a median number of 9 repeats per lincRNAs. In the Methods section, I read that for the high-throughput microscopy and flow cytometry for cell-size and cell-cycle phenotypes, over 80% of the 110 lincRNA mutants screened for cellular phenotypes were assayed in at least 2 independent biological repeats. For the overexpression, I read that each strain was represented by at least 12 colonies across 3 different plates and experiments were repeated at least 3 times. Each condition was assayed in three independent biological repeats, together with control EMM2 plates, resulting in at least 36 data points per strain per condition.Perhaps I missed something. If not, could the authors clarify this? In addition, I suggest to indicate the number of replicates used for each lincRNA/condition/assay in Supplemental Dataset 2 (I could only find the information for the Flow Cytometry) and in Supplemental Dataset 6.

For all colony-based phenotyping, we performed at least three biological repeats, meaning that the strains were assayed three times independently, each with at least two colonies (technical repeats). In most cases, we had two or more independently generated deletion strains for each lincRNA, and we performed at least three biological repeats for each strain (hence the higher median number of nine repeats per lincRNA). The numbers of independent deletion strains for each lincRNA are provided in Supplemental Dataset 1 (sheet: lincRNA_metadata, column: n_independent_ko_mutants). The total numbers of repeats carried out for each condition after QC filtering are available in Supplemental Dataset 2 (columns: observation_count). We have now clarified this on p. 6, and the details are provided in the Methods on p. 28-29 (for deletion mutants) and p. 32 (for overexpression mutants). For the high-throughput microscopy and flow cytometry experiments, we performed the repeats as described in the text.

Minor comments:1. Specific experimental issues that are easily addressable.– The pattern of the SPNCRNA.1343 and SPNCRNA.989 mutants is consistent with the idea that these lincRNAs act in cis and that their deletion interferes with the expression of the adjacent tgp1 and atd1 genes, respectively. The authors could easily test by RT-qPCR or Northern Blot that the lincRNA deletion leads to the induction of the adjacent gene. Also, if the hypothesis of the authors is correct, the ectopic expression of these two lincRNAs in trans should not complement the phenotypes of the corresponding mutants. These experiments would reinforce the conclusion of the authors about the specific regulatory effect of the SPNCRNA.1343 and SPNCRNA.989 lincRNAs.

It would actually not be as easy as suggested to obtain conclusive results in this respect. For *SPNCRNA.1343* and its neighbour, *atd1,* the mechanisms involved have already been shown in detail based on several mechanistic studies (Ard et al., 2014; Ard and Allshire, 2016; Garg et al., 2018; Shah et al., 2014; 2014; Yague-Sanz et al., 2020). But these studies did require multiple precise genetic constructs and specialized approaches to interrogate the complex regulatory relationships between the overlapping transcripts which can be both positive and negative. As correctly pointed out by Reviewer 2, we do not know the particular conditions where any cis-regulatory interactions take place, and a negative result would not be conclusive. We have interrogated our RNA-seq data obtained under multiple genetic and environmental conditions (Atkinson et al. 2018) to analyse the regulatory relationship between *SPNCRNA.1343* and *atd1* (studied before) as well as *SPNCRNA.989* and *tgp1* (proposed in our manuscript). Depending on the specific conditions, both of these gene pairs show positive or negative correlations in expression levels. So it is not possible to just perform the easy experiment as suggested to reach a clear conclusion.

– Is there any possibility that some nutrient/drug/stress conditions interfere with the expression from the nmt1 promoter?

This seems unlikely as this widely used promoter is known to be specifically regulated by thiamine. Consistent with this, we actually detected phenotypes for over 90% of the overexpression strains. But we cannot exclude the possibility that some conditions might interfere with *nmt1* function.

– Supplemental Figure 7 refers to unpublished data from Maria Rodriguez-Lopez. Is this still allowed?

These are just control RNA-seq data from wild-type cells growing in rich medium. It does not seem that meaningful, but if required we could submit these data to the European Nucleotide Archive (ENA).

– Supplemental Figure 8 shows drop assays to validate the growth phenotypes revealed by the screen for lincRNAs of clusters 1 and 3. As admitted by the authors in the text, in most cases, the effects are quite difficult to see to the naked eye. Did the authors consider the possibility to use growth curves (for the lincRNAs/conditions they would like to highlight), which might be more appropriate to visualize weak effects?

We have tried a few experiments in liquid medium using our BioLector microfermentor. However, the doses need to be substantially changed for liquid media (in which cells typically are more sensitive than on solid media). So the situation with the altered conditions would become too confusing and could not be used as a direct validation of our results from solid media.

2. Are prior studies referenced appropriately?Yes. The authors could have cited the work of Huber et al. (2016) Cell Rep. (PMID: 27292640) as another pioneer study where systematic lncRNA deletion was performed, even if in this case, these were antisense lncRNAs.

Agreed, we now cite this paper in the Introduction (p. 4).

3. Are the text and figures clear and accurate?Overall, I found the text and figures clear.Reviewer #3 (Significance (Required)):Eukaryotic genomes produce thousands of long non-coding RNAs, including lincRNAs which are expressed from intergenic regions and do not overlap PCGs. Several lincRNAs have been extensively studied and characterized, showing that they function in different cellular processes, such as regulation of gene expression, chromatin modification, etc. However, beside these well documented lincRNAs, the function of most lincRNAs remains elusive. In addition, under the standard growth conditions used in labs, many of them are expressed to very low levels, and for the few cases for which it has been tested, the deletion and/or overexpression in trans often failed to display in a detectable phenotype.High throughput approaches for lncRNA functional profiling are currently emerging. The lab of Jurg Bahler recently developed a high throughput colony-based screen approach enabling them to quantitatively assay the growth and viability of fission yeast mutants under multiple conditions (Kamrad et al., eLife 2020). Here, they take advantage of this approach to characterize mutants of 150 lincRNAs in fission yeast, including not only deletion mutants generated using the CRISPR/Cas9 technology, but also overexpression mutants, tested in 149 and 47 growth conditions, respectively. This systematic approach allowed the authors to reveal specific phenotypes for a large fraction of the lincRNAs, emphasizing the fact that they are likely to be functional in particular nutrient/drug/stress conditions, acting in cis but also in trans.As I wrote in the summary above, I think that this study is important and constitutes a significant contribution in the lncRNA field.My field of expertise: long non-coding RNAs, yeast, genetics.Referee Cross-commenting:I can see that reviewer #1 and I have raised the same concerns about the lack of insert sequencing for the overexpression plasmids, which is crucial to control that the correct lincRNAs were cloned and that no mutation has been introduced by the PCR. We are also both asking for RT-qPCR controls to show that the lincRNAs are indeed overexpressed. Again, this control is very important as many long non-coding RNAs are rapidly degraded by the nuclear and/or ctyoplasmic RNA decay machineries. So expressing a lincRNA from a plasmid, under the control of a strong promoter, does not guarantee increased RNA levels.I see that reviewer #2 is asking for a gametogenesis assay. I think it should be limited to the 3 lincRNAs which belong to the same sub-cluster as meiRNA.